

# Evaluating the Efficiency of Subsurface Drainages for Li-Shan Landslide in Taiwan
Der-Guey Lin[1], Sheng-Hsiung Hung[2], Cheng-Yu Ku[3], Hsun-Chuan Chan[4]*
[1]Professor, Department of Soil and Water Conservation, National Chung-Hsing University
[2]Doctoral student, Department of Soil and Water Conservation, National Chung-Hsing University
[3]Professor, Department of Harbor and River Engineering, National Taiwan Ocean University
[4]Associate Professor, Department of Soil and Water Conservation, National Chung-Hsing University
[4]hcchan@nchu.edu.tw (*corresponding author: No. 250 Kuo-Kuang Road, Taichung 402, Taiwan)
**Abstract:** This study investigates the efficiency of subsurface drainage systems includes drainage wells (vertical shaft with
drainage boreholes or horizontal drains) and drainage galleries (longitudinal tunnel with sub-vertical drainage boreholes) for the
slope stabilization of Li-Shan landslide in central Taiwan. The efficiency of the subsurface drainages is verified through a series
of two-dimensional (2-D) rainfall induced seepage and slope stability analyses without and with subsurface drainages
remediation during two typhoon events. Numerical results and monitoring data both show that the groundwater level at *B5*
monitoring station with subsurface drainages remediation during Toraji Typhoon (2001) is about 40 m lower than that without
remediation during Amber Typhoon (1997), and the factor of safety *Fs* of the first potential sliding surface (*1st-PSS*, the most
critical potential sliding surface) is promoted simultaneously from 1.096 to 1.228 due to the function of subsurface drainage
systems. In addition, the *Fs* values of the three potential sliding surfaces (*1st-PSS*, *2nd-PSS*, and *3rd-PSS*) stabilized by subsurface
drainage systems are constantly maintained greater than unity ($F_S > 1.0$ or $F_S \geq 1.217$) during rainfalls with return periods
increases from 25 to 50 and 100 years. This demonstrates the subsurface drainage systems in Li-Shan landslide are functional
and capable of accelerating the drainage of infiltration rainwater induced from high intensity and long duration rainfall and
protect the slope of landslide from further deterioration.
**Keywords:** landslides, subsurface drainage systems, drainage boreholes, drainage well, drainage gallery, potential sliding
surface, factor of safety
## 1.    Introduction

The Li-Shan landslide, a large scale landslide on the mountainous area of central Taiwan, currently has been stabilized by
the subsurface drainage systems consisted of drain wells and drainage galleries. The landside has a long history of intermittent
large movements toward down slope during rainfall. In April 1990, a long duration torrential rainfall triggered a massive
landslide in Li-Shan area where immediately located at the intersection of two East-West cross-island highways, namely, Routes
Tai-8 and Tai-7. The catastrophic event caused large ground movements and severe damages on Route Tai-7 and Li-Shan Hotel
in the southeast region of the landslide and the hotel is one of the Guest Houses of past president Chiang Kai-Shek, landmark
architecture in Li-Shan area. After the disastrous event, to prevent the expansion of the landslide, the relevant public agencies
approved an emergency plan entitled "Investigation and Remediation Planning for Landslides in Li-Shan Area" for three years'
duration from 1991 to 1993 to implement a comprehensive field investigation and engineering design for the landslide.
Subsequently, on June 25, 1994, Taiwan government starts to execute an emergency plan called "Remediation Plan for Li-Shan
Landslide" for seven years' duration from 1995 to 2002 to cope with the complicated and unfavorable hydrological and
geological situations of the landslide. The main remediation work for Li-Shan landslide is to lower the groundwater level
through different subsurface drainage systems during the rainfall of typhoon seasons.
### 1.1 Location and Development of Li-Shan Landslide Area
For administrative district, the Li-Shan landslide area (or Li-Shan landslide) comes within the jurisdiction of Li-Shan
village, Taichung City Government, Taiwan and has a population around 2000. Li-Shan landslide is situated in Central
Mountains at the northeast of Taichung City with a distance about 100 km and also at the intersection of Route Tai-8 and Route
Tai-7 of the East-West cross-island highways where locates the landmark architecture Li-Shan Hotel as shown in Fig. 1 (a).
Because of the location, Li-Shan village eventually becomes a key spot to synthesize the East-West transportation,
commercial business, sightseeing and tourism of central Taiwan. During 1970's and 1980's, a vast area of primary forest was
cultivated into orchard and a great quantity of fruiters, vegetables and high economic crops such as tea trees were planted in
Li-Shan as displayed in Fig. 1(b). As a consequence, those agricultural activities enrich local resident, however, damage the
environment due to improper soil and water conservation.



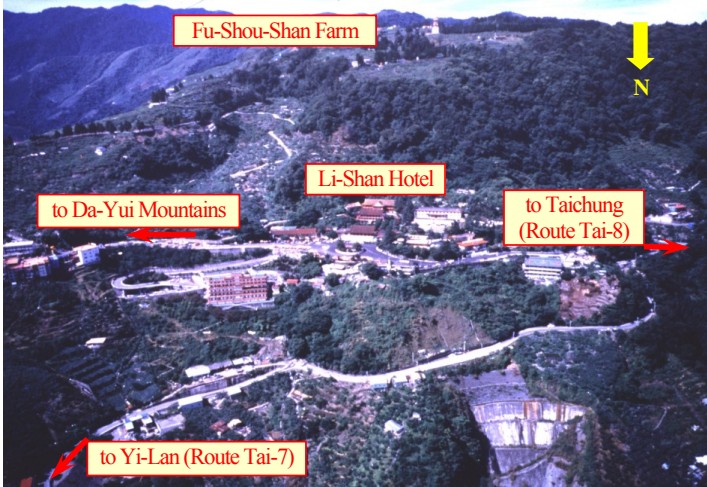

(a)

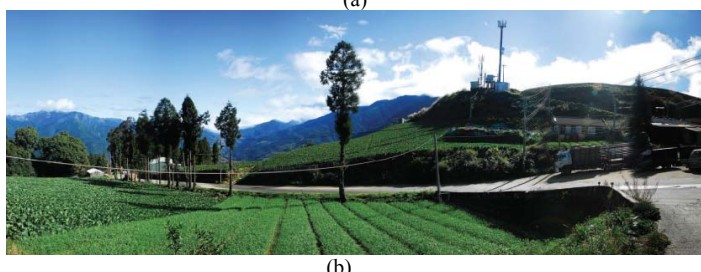

(b)
Fig. 1 (a) Overlook Li-Shan landslide area northward from Fu-Shou-Shan Farm at the upslope
(b) Enormous agricultural cultivation with high economic crops

**1.2 Climate**


The temperature in Li-Shan varies greatly between the day and the night and the temperature is about 15.2 °C on an average.
In Li-Shan the average annual rainfall approximates 2,242 mm for an average annual rainy day of 176 days based on the rainfall
records from 1978 to 2008. Annually, most of rainfall concentrates on Spring and Summer (or from March to September) and in
May and June the average monthly rainfall can reach 514 mm. In addition, the torrential rainfall occurred 7 or 8 times annually
with rainfall intensity of 100 mm/day during June and September. However, from October to next February, the weather turns
into a dry season and the rainfall in this duration is merely 20.2% of average annual rainfall. Conclusively, the rainfall of
Li-Shan is mainly influenced by the mould rains season (or plum rains season) and its topography.

**1.3 Topography and Geology**


As shown in Fig. 2, the Li-Shan landslide situates at the west of Central Mountains with an area of around 230 hectares and
it looks similar to a reverse triangular shape from south to north. The terrain of the landslide is descending from south to north
with elevation varying from 2,100 to 1,800 m. The landslide is characterized by hilly and valley topography and the Da-Jai River
flow from east to west through the northern edge of the landslide. Topographically, Li-Shan landside is situated in the valley of
the Da-Jia River and classified as an old ancient landslide. There is an old sliding body located at the center of the landside and
a smaller sliding body can be identified by field investigations as well.
The Li-Shan fault, a major ridge fault of Taiwan Island generated by the tectonic activity of the westward thrust front due to
the collision between the Philippine Sea Plate and the Eurasian Plate, just locates at few kilometers west of the Li-Shan landslide.
As geological heterogeneity is generally recognized as a crucial factor in rainfall-induced seepage and slope stability analyses,
the evaluation of the efficiency of subsurface drainage systems should take the complexity of the soil strata into account. The
geology of the landslide is categorized into Miocene Lu-Shan formation, highly fragmentary tertiary sub-metamorphic rock, and
thick colluvium encountered locally and occasionally mixed with mudstone enriched with cleavage. In this region, through the
field data of boring log and geophysical exploration, the soil strata can be classified into five types from shallows to depths based
on their weathering degree as shown in Fig. 3(a), namely, (1) colluvium, (2) heavily-weathered slate, (3) medium-weathered
slate, (4) lightly-weathered slate, and (5) fresh slate. The material features of the five types were also evaluated by the *ISRM*
classification as listed in Table 1 and it can be verified that the maximum weathering depth approximates 63 m at least. The
landslide area can be divided into three regions, i.e. the west, northeast, and southeast regions. Except the southeast region, most
of the unstable slopes possess shallow sliding planes about 9~26 m below ground surface. However, there is an old landslide
within the southeast region. According to the core logs and the records of drainage gallery construction, the old sliding surface is
located more than 40~60 m below ground surface.




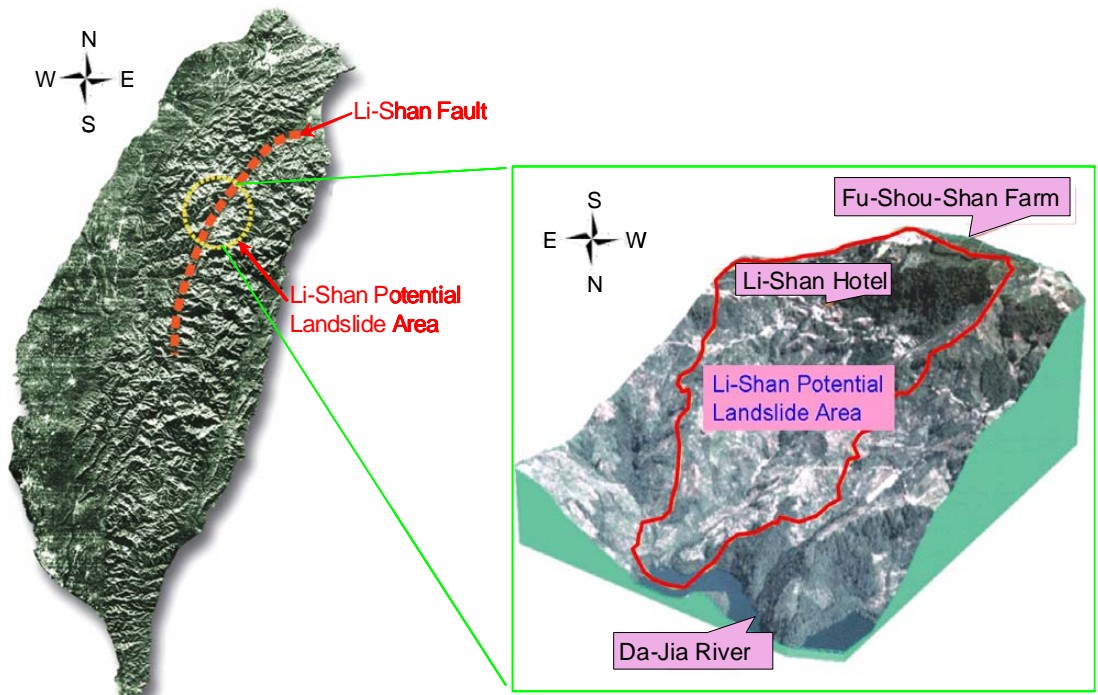

Fig. 2 Topographic and geological characteristics of Li-Shan landslide located at central Taiwan

Conclusively, the potential sliding surfaces are basically along the lower boundary of the regolith. The slide is mainly made
up of the colluviums and heavily-weathered slate and forming the main part of the Li-Shan landslide. The outcrops of the
Li-Shan landslide can be categorized into Miocene Lu-Shan formation and mainly consist of slate with color varied from black
to deep gray as shown in Fig. 3(b). Nevertheless, the sliding bodies overlying the potential sliding surfaces of the landslide
primarily is composed of weathered slate, fragment of slate and intercalary clayey strata. Conclusively, the properties of the
sliding bodies exhibit a loose texture and poor grain size distribution which alternately leads to a less cementation, low strength,
and high permeability geo-material. In addition, the composition of fresh slate can be visualized by microscopic image as
displayed in Fig. 3(c).

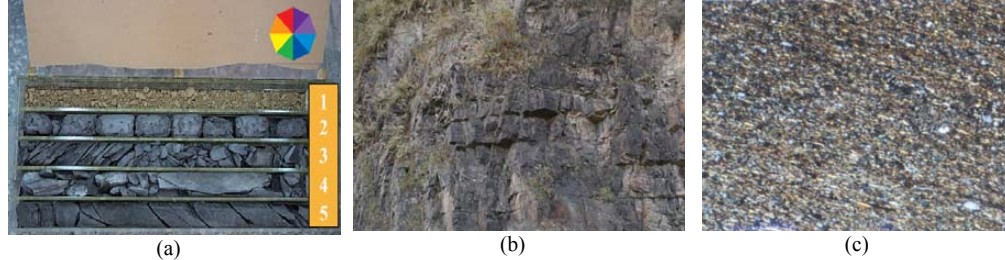

       (a)             (b)             (c)

Fig. 3 (a) material types of soil strata (b) outcrops of slate formation (c) microscopic image of fresh slate of Li-Shan landslide

Table 1 Features of soil strata for Li-Shan landslide

| Material (sampling depth: m) | Descriptions | ISRM |
|---|---|---|
| 1. Colluvium (-1 m) | Sandy silt of yellowish-brown color mixed with rock fragments and gravel | VI |
| 2. Heavily-weathered slate (-13 m) | Clayey soil, silty sand or sandy soil of black color with texture similar to fresh rock | V |
| 3. Medium-weathered slate (-23 m) | Fragmentary rock core with thin sheet, black color, grain size of 2~30 mm and the outcrop enriched with fissure. | III, IV |
| 4. Lightly-weathered slate (-18 m) | Blocky rock core with rounded shape, black color, grain size of 5~10 mm and the outcrop similar to fresh rock | II |
| 5. Fresh slate (-63 m) | Cylindrical rock core with black color, length>50 mm, and $RQD$>75 | I |





**1.4 Landslide in 1990**

In the past years, ground movements frequently occurred at the landslide area during seasonal and typhoon rainfalls. In particular, the features of topography, geology, meteorology, hydrology, and poor drainage of Li-Shan landslide area cause the slope lands in this area liable to situate in an unfavorable conditions for stability. Due to a series vast agricultural cultivation during 1970′s and 1980′s, a consecutive five days torrential rainfall from 11th to 15th, April in 1990 triggered a massive landslide and damage the Route Tai-7 of central cross-island highway which completely interrupted the transportation of east/west direction. The landslide extends over a length of 150 m oriented SE to NW with a width of 100 m and a mean slope of 20°. The total volume of the sliding mass is about $3{\times}10^5$ m$^3$ (or 0.3×million m$^3$) with an average thickness of 20 m. In addition, at the waist part of the landslide, groundwater flows out with a discharge rate of around 900 liter/min. As presented in Fig. 4, the catastrophic landslide event causes a severe depression on the sightseeing industry of Li-Shan and the Li-Shan Hotel was also closed down due to the detrimental subsidence induced from the landslide.

Because of the high erosion rate of the Lu-Shan Formation, together with the heavy rainfall during April 15~19, 1990, it is generally concluded that the landslide is predominantly caused by the infiltrated rainwater and poor drainage condition. Although the study area locates at the southeast region of the landslide, at the upslope of Li-Shan Hotel, as shown in Figs.1 and 2, the entire slope land was cultivated into Fu-Shou-Shan Farm for agricultural plantation purpose. As a consequence, in addition to direct surface infiltration, water from the irrigation system of the farm enters the landslide area was also inferred to be a crucial factor triggering the landslide.

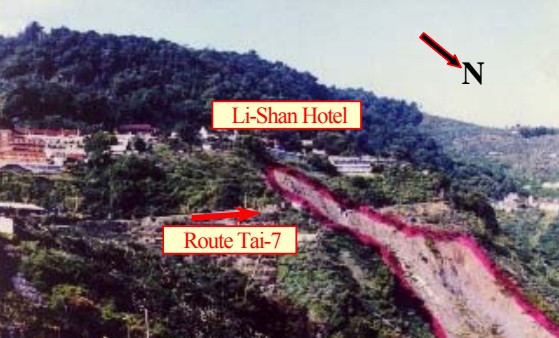
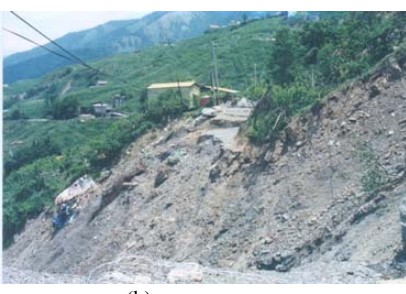

(a)                                                      (b)

Fig. 4 Li-Shan landslide on April, 15~19, 1990 (a) sliding mass moves from south to north direction (b) foundation failure of Route Tai-7

**1.5 Analysis of Rainfall Records for Landslide in 1990**

As illustrated in Fig. 5, a maximum daily rainfall of 155.5 mm occurred on 19th April, 1990 with occurrence frequency of 1.87 years and it is not heavy for a daily rainfall. Nevertheless, the cumulative rainfall for the periods of 10th~20th April, 1990 approximates 586 mm, meanwhile the total cumulative rainfall for the entire April in 1990 can reach 957.5 mm. These records are maxima with occurrence frequency higher than 50 years when compared with the records of rainfall events in the past.

In addition to the influences of topography and geology, landslide occurs frequently in Li-Shan area due to large amounts of rainwater in rainy season and torrential rainfall in typhoon season. As a consequence, massive infiltrated rainwater induced from the consecutive rainfall and stored up in the sliding body will eventually turned into a crucial factor to trigger a large scale landslide. The infiltrated rainwater in the upslope of the sliding body will seep downwards and accumulate to raise the groundwater level and it alternately increases the pore-water pressure on the potential sliding surface of sliding body. Consequently, the sliding failures of colluviums and weathered slate in this region (southeast region) can be attributed to the infiltration of rainwater and rise of groundwater level.

**1.6 Implementation of Remediation for Li-Shan Landslide**

After the large scale landslide event in 1990, the field observations showed that the scope and scale of the landslide were constantly expanding. According to the site investigations on the distribution of sliding bodies within the landslide area from 1990 to 2008, it was found that the scope influenced by sliding bodies and slope failure are exceptionally extensive as shown in Fig. 6. The potential sliding surface of Li-Shan landslide is deep-seated approximately at a depth of 30~70 m and spreads in a large area. The overburden above potential sliding surface mainly consists of colluviums and weathered slate with high permeability.




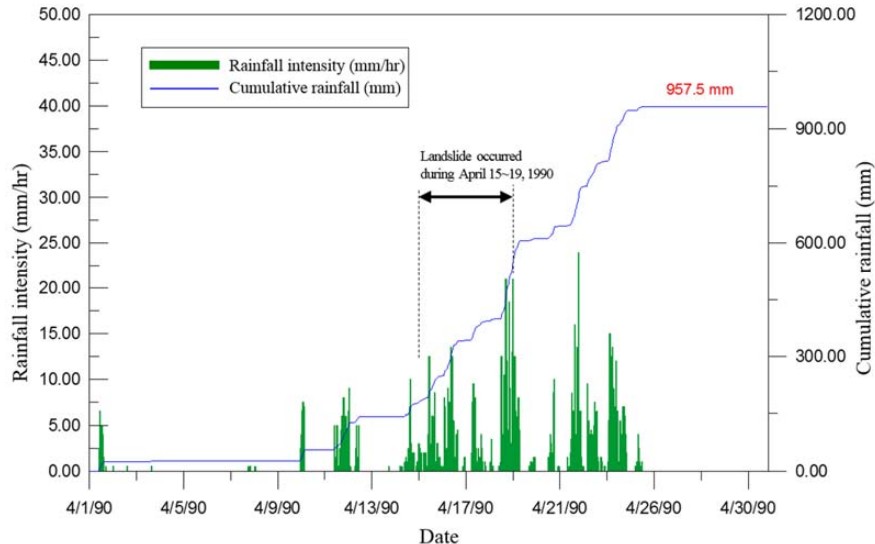

Fig. 5 Precipitation record of April in 1990 from Li-Shan rainfall monitoring station

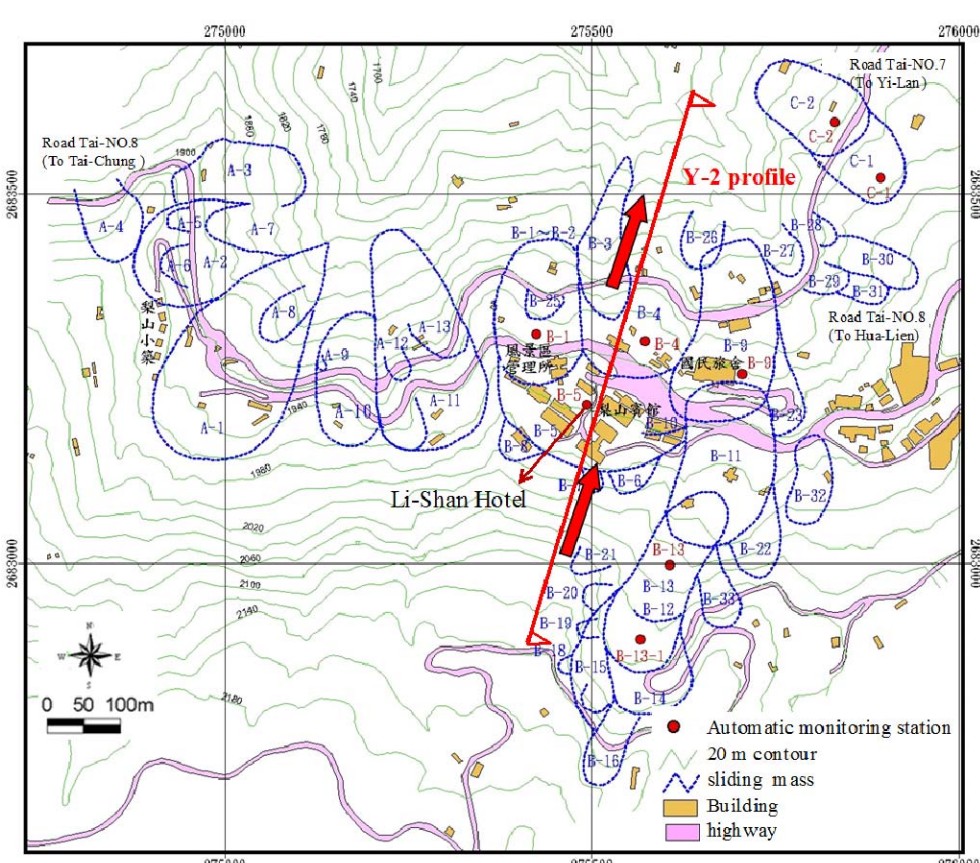

Fig. 6 Distribution of sliding bodies at the southeast region of Li-Shan landslide and the arrow along the *Y-2 profile* line denotes the movement direction
of the landslide (SWCB, 2008)

On June 25, 1994, a seven years' remediation plan from 1995 to 2002 with total expenditure about 0.915 billion NT$ was
officially approved by Taiwan government which threw all positive factors such as manpower, financial and material resources
into the remediation works to mitigate the spread potential of the landslide. However, it was the first time in Taiwan to perform
such enormous plan for landslide remediation and it might be also rare in the case history of slope remediation elsewhere. The
remediation plan encompassed drainage galleries, drainage wells (vertical shaft with radial drainage boreholes drilled at


multi-elevation inside the shaft), drainage boreholes at slope toe (subsurface drainage with shallow depth), submerged dam (for
erosion control) and check dam (for sediment control), as shown in Fig. 7, were implemented to improve the slope stability of
Li-Shan landslide.
168       According to the field investigation and the analyses of remediation plan, the factor of safety for several main sliding
bodies in Li-Shan landslide was in a range of 0.98~1.10 (=$F_S$) which represents the slopes situated on the verge of critical state or
creep condition and needs remediation works to promote the stability. Securing the safety of local residences and recovering the
traffic transportation of Route Tai-7 were the major objectives of the remediation plan. The primary works of remediation plan
was to expedite the drainage of infiltrated rainwater and lower the groundwater level. It was estimated that the factor safety of
sliding bodies could be promoted up to $F_S$ =1.2 for a groundwater level approximately drawn down for 8.5 m (SWCB, 2003). In
the collapsed and sliding zone, some restrain engineering works and slope protections were also installed to ensure the slope
stability and resume the traffic transportation of Route Tai-7. Conclusively, the main remediation works of emergency for
Li-Shan landslide were the subsurface drainage systems which consists of 15 drainage wells installed from 1995 to 2000, and 2
drainage galleries (*G1*- and *G2*-gallery) constructed from 1997 to 2002.

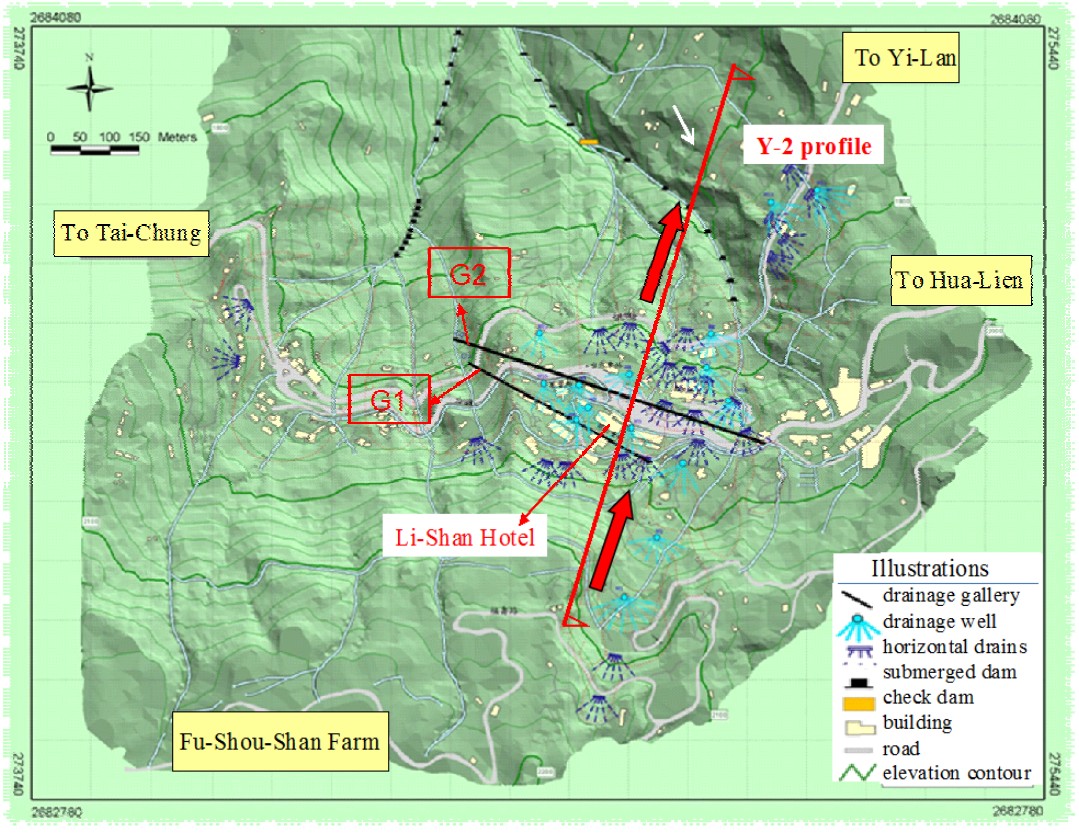

180                 Fig. 7 Configurations of subsurface drainages and remediation works
181                       in Li-Shan landslide (2008,SWCB)
**2. Subsurface Drainage Systems in Li-Shan Landslides**
185       Li-Shan landslides were frequently triggered by a rise of groundwater level accompanied with increasing pore-water
pressure on potential sliding surface. Accelerating and improving subsurface drainage can stabilize a large volume of sliding
body at comparatively low engineering cost and it can be a very attractive option for many landslides remediation. As a result,
drainage is by far the most commonly used methods for stabilizing large scale unstable slopes, either alone or in conjunction
with other method in Taiwan. Attempts have been made to provide a design method to optimize the number and spacing of
horizontal drains (or drainage boreholes) (Kenney et al., 1977; Prellwitz, 1978; Long, 1986). All methods are based on
groundwater flow principles and the major difficulty with theoretical design methods is that the permeability is assumed to be
constant throughout the ground. Xanthakos et al. (1994) indicated that natural slopes are rarely homogeneous enough to allow
reliable subsurface drainage design according to simple principles of dewatering. In addition, Hausmann (1992) suggested that
for a successful dewatering system, the designer must have a good understanding of geological structures and choose a drainage
system layout that increases the probability of intersecting the major water-bearing stratum. The effectiveness of horizontal
drainage system was investigated by Rahardjo et al. (2002, 2003) through a series of numerical analyses on the location and
length of horizontal drains (or drainage boreholes). It was found that the horizontal drain is effective in lowering the
groundwater table and most effective when located at the bottom zone of a slope.



In such circumstances, for the design of subsurface drainage systems in Li-Shan landslide, the installation locations of
drainage wells and drainage galleries accompanied with well-configured drainage boreholes (or horizontal drains) become
extremely crucial to the efficiency of subsurface drainage systems.
**2.1 Drainage Well (Vertical Shaft with Drainage Boreholes)**
The drainage well in Li-Shan landslides, which consists of vertical shaft, drainage boreholes (or horizontal drains), stilling
pond, and drainage pipe, is a very effective working method to remove the confined groundwater in soil strata and the method
was mainly adopted to get rid of the groundwater situating at large depth as illustrated in Fig. 8(a). Large amounts of water can
be drained from the slope through drainage wells accompanied with a consequent drop of groundwater levels. Up to the present,
there totally 15 drainage wells (1995~2000) were installed in Li-Shan landslides as shown in Fig. 8(b).

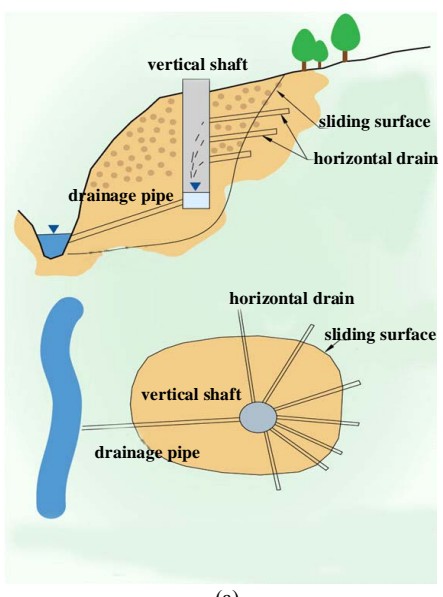
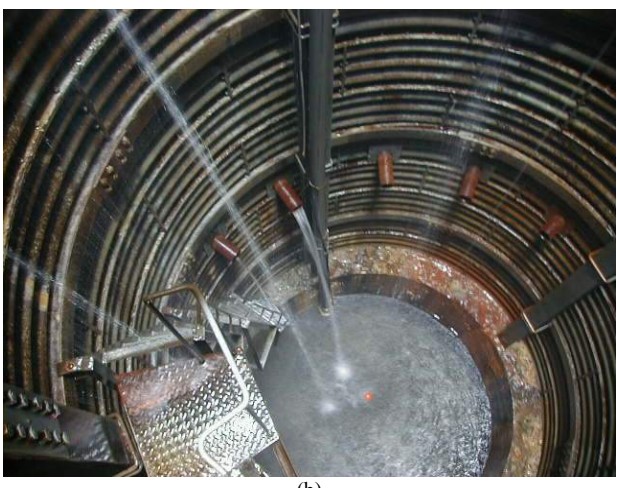

(a)                         (b)

Fig. 8 (a) configuration of vertical shaft with three-level of drainage boreholes (or horizontal drains) in landslide (b) vertical shaft assembled by
corrugated steel sheets and collecting groundwater through drainage boreholes (SWCB, 2003)

  The vertical shaft was assembled by a continuous galvanized corrugated steel sheet liner with a diameter of 3.5 m and
penetration depth of 15~40 m to reach deep-seated potential sliding surfaces. By installing a vertical shaft near the upper portion
of sliding body, an array of 5~10 uncased drainage boreholes (spacing about 1~2.5 m) with a diameter of 70~100 mm and length
of 40~70 m, radiating from the interior of vertical shaft, were drilled at 3 different elevations and inclined 2°~10° (typically 5° to
horizontal) upward into the upslope of sliding body. Comparatively, Matti, et. al, (2012) indicated a mean spacing between the
drainage boreholes of 10 m is sufficient to control the temporal head fluctuations between the wells within a range of a few
meters. Subsequently, a 50 mm diameter perforated *PVC* pipe wrapped in filter fabric was fitted into the drainage borehole
(becomes horizontal drain) to intercept the downwards seeping groundwater flow by gravity. A concrete stilling pond (or
storage pond) with depth of 1.0~1.5 m and slab thickness of 50 cm was constructed at the bottom of the shaft using water-tight
concrete to accumulate the groundwater from drainage boreholes and eventually discharge to the existing drainage system at a
lower elevation than the shaft base by gravity through a *PVC* or *HDPE* pipe with a diameter of 100 mm and an inclination of 3°
～5° to horizontal.

  In this study, the *Y2-profile* of Li-Shan landslide was adopted for seepage and stability analyses as shown in Figs. 6 and 7,
and three drainage wells *W6*, *W7* and *W8* with a penetration depth of 20, 25, and 15 m respectively were installed adjacent to the
*Y2-profile*.

**2.2 Drainage Galleries with Sub-vertical Drainage Boreholes**
The groundwater level variation after installing 7 drainage wells (1995~1997) in Li-Shan landslide indicated that to
entirely drain off the infiltrated rainwater at a great depth remains difficult and impractical. In such situations, a decision was
made to construct two drainage galleries (1997~2003) to dewater the sliding bodies of large volume instead of requiring a
substantial number of drainage boreholes when groundwater level is deep-seated and impossible to reach by drainage wells. In
Li-Shan landslide drainage gallery serves to lower the general groundwater within the landslide mass and to tap into a specific
area of high permeability or aquifer at the upper reach of the landslide so that groundwater levels are further reduced.
As shown in Fig. 7, at present two drainage galleries totaled about 900 m in length (*G1*-gallery=350 m, 1999~2001;
*G2*-gallery=550 m, 1997~2003) passed through the *Y-2 profile* at the southeast region of Li-Shan landslide. The gallery portals
were located at an elevation of 1,910 m and 1,865 m a.s.l. for *G1*- and *G2*-gallery respectively and the galleries were then


excavated from northwest to southeast by an upward grade of 1~2 % to facilitate drainage, as illustrated in Fig. 9. Along the
gallery several water-collection chambers with a fan-shaped array of sub-vertical drainage boreholes were drilled to lower the
groundwater level under the Li-Shan Hotel. Groundwater is intercepted and evacuated from the potential sliding surface of
landslide by gravity through a network of drainage boreholes connected to the water-collection chamber of drainage gallery
situated below the potential sliding surface of the landslide. Due to the fact that the depth of potential sliding surfaces of Li-Shan
landslide ranges from 30 to 70 m, the drainage galleries were decisively constructed within the intact stable fresh bedrock about
80 m deep underlain the unstable colluviums and weathered bedrock. Eventually the drainage galleries would not influenced by
the landslide movements.
As shown in Figs. 10(a) and (b), the gallery has a smaller dimension of 2.07 m×2.1 m (=height×width) with a horseshoe
shape cross section and semi-circle crown. Galvanized corrugated steel liner was used for the lateral support of gallery and
water-tight concrete drainage ditch was constructed at the base of gallery to drain off the groundwater from the water-collection
chambers.

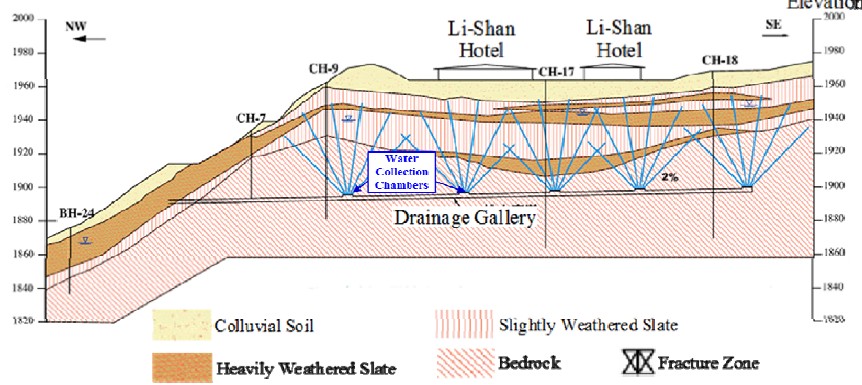

Fig. 9 Geological cross section and location of the drainage gallery (*G1*-gallery)

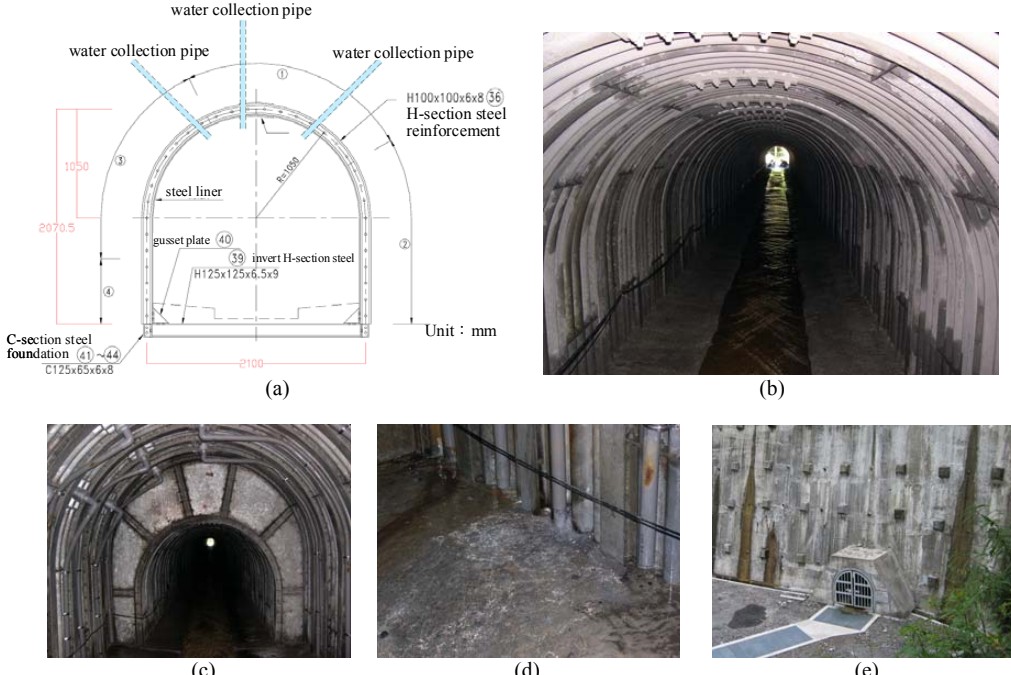

Fig. 10 (a) (b) gallery with a smaller cross sectional layout (width×height=2.07 m×2.1 m) for drainage (c) water-collection chamber with enlarged cross
section (width×height=3 m×3 m) and sub-vertical drainage boreholes (water collection pipes) (d) groundwater collected in chamber (e) outlet of gallery

As displayed in Figs. 9 and 10(c), 5 water-collection chambers were set up along the *G1*-gallery (10 water-collection
chambers along the *G2*-gallery) and each chamber is 6 m long and has an enlarged cross section of 3.0 m×3.0 m
(=height×width). Meanwhile, for each chamber there was 18 (=3×6) sub-vertical drainage boreholes (or water collection pipes)




with a length of 40~60 m were drilled upwards at the crown of gallery to collect and drain off the groundwater from upper soil strata. As shown in Figs. 9 and 10(a), the average spacing of sub-vertical drainage borehole in a chamber approximates 1.0 m×1.0 m (=transverse spacing×longitudinal spacing). As a result, there 90 drainage boreholes (=5×18) with a total length of 4,873 m were drilled for *G1*-gallery (180 drainage boreholes (=10×18) and 10,700 m long for *G2*-gallery). In addition, according to the monitoring data, the drainage galleries can intercept and drain the groundwater from the sliding bodies by a flow rate $Q$ ranged from 36 to 90 m$^3$/hr (for *G1*-gallery $Q_{G1}$= 60~90 m$^3$/hr, or *G2*-gallery $Q_{G2}$=36~60 m$^3$/hr).

Although the efficiency of the drainage gallery to stabilize unstable slopes has been studied in a number of case histories by some researchers (Eberhardt et al., 2007; Matti, et. al, 2012), the functional performance and efficiency of subsurface drainage systems constructed in Li-Shan landside with relatively high construction costs (0.915 billion NT$) has not yet been evaluated during torrential rainfall. In particular, the effects of the two drainage galleries (*G1*- and *G2*-gallery) on the slope stability of Li-Shan landslide during rainfall (or specific crisis) have not been inspected up-to-date. Using monitoring data and numerical techniques this study takes the effect of rainwater infiltration into account during typhoons to verify the function of subsurface drainages to stabilize the landslide quantitatively.

## 3.  Methodology

The numerical model of *Y2-profile* was established according to the topography, hydrology and subsurface drainage remediation in Li-Shan landslide. Rainfall-induced seepage analyses and slope stability analyses before and after subsurface drainages remediation were carried out using finite element method (*FEM*) and limit equilibrium method (*LEM*). The *FEM* seepage analyses involves calculating the pore-water pressure field throughout the problem domain, which is then introduced along the potential sliding surface for each time step into the *LEM* stability analyses. These two-dimensional (2-D) numerical models evaluate the efficiency of the drainage wells and drainage galleries installed within and below the sliding bodies with the aim of lowering the groundwater levels and promoting the factor of safety of the landslide. It should be noted that this study concentrates on the transient seepage modeling rather than the deformation analysis because one of the purposes is to demonstrate how to integrate transient seepage modeling into the stability analysis of intricate landslide. Based on the variations of groundwater levels, volumetric water content and factor of safety of the potential sliding bodies, one can recognize the effects of rainfall-induced seepage and subsurface drainages on the slope stability of Li-Shan landslide. The flow chart of working procedure for the study was illustrated in Fig. 11.

### 3.1  Initial and Boundary Conditions

The *Y2-profile* situates at the southeast region of Li-Shan landslide and passes through the *B4* and *B5* sliding bodies, as shown in Figs. 6 and 7, was selected as a representative profile for numerical analyses. In the analyses, the soil strata were simplified in sequence from ground surface to underground as: colluviums, heavily to medium weathered slate, and slightly weathered to intact bedrock. The numerical model of geological profile is illustrated as Fig. 12 and a key element in the model is to incorporate the subsurface drainage systems into the simulations. The elevations of left and right boundary of the model are 2,156 and 1,768 m, respectively and the distance of bottom boundary extended from left to right is 830 m.

Rainfall-induced seepage analyses consist of steady and transient analyses. For steady analysis, the initial groundwater level and distributions of pore-water pressure prior to a main rainfall event were generated by assigning a constant total head at the left and right boundaries of the model and which alternatively used as initial boundary conditions for the sequential transient analysis to calculate the time dependent groundwater level and slope stability. Based on the parametric analyses, Ng, CWW and Shi, Q (1998) evidently indicated that the initial groundwater condition prior to the rainfall has a significant effect on the slope stability. In this study, incorporating continuous measurements of groundwater levels from observation wells with the left and right constant total head boundaries, one can determine the average initial groundwater level for an ordinary time. For transient analysis, the groundwater level and pore-water pressure calculated for the last time step ($t_{i-1}$) were sequentially used as the initial condition of  the seepage and stability analyses for the current time step ($t_i$).

Due to the complexity of the general geology of the landslide, simplifications are made in the transient seepage and stability numerical models. As shown in Fig. 12(a), in numerical model, the *AB* ground surface boundary was specified as a rainfall infiltration boundary, while the *CD* bottom boundary was defined as an impermeable close boundary without seepage (discharge rate $Q$=0). In addition, according to the monitoring  data of groundwater levels prior to a rainfall event, the *AD* left boundary and *BC* right boundary were assigned as constant head boundaries with total heads $H$= 2,140 and 1,750 m respectively. The finite element mesh of numerical model encompassed drainage wells *W-6*, *W-7*, *W-8*, and *H-10*; groundwater level observation wells *B4*, *B5* and drainage galleries *G1*, *G2* located along *Y2-profile* are illustrated in Fig. 12 (b). In addition, it can be found that the subsurface drainage systems were mainly installed at the region of the middle crest or the middle platform of the slope to cope with a large amount of rainwater infiltration during torrential rainfall. This coincides with the numerical results presented by Gasmo J. M. et al. (2000) which reveals that most of infiltration occurs at the crest (or a flat platform) of a slope.



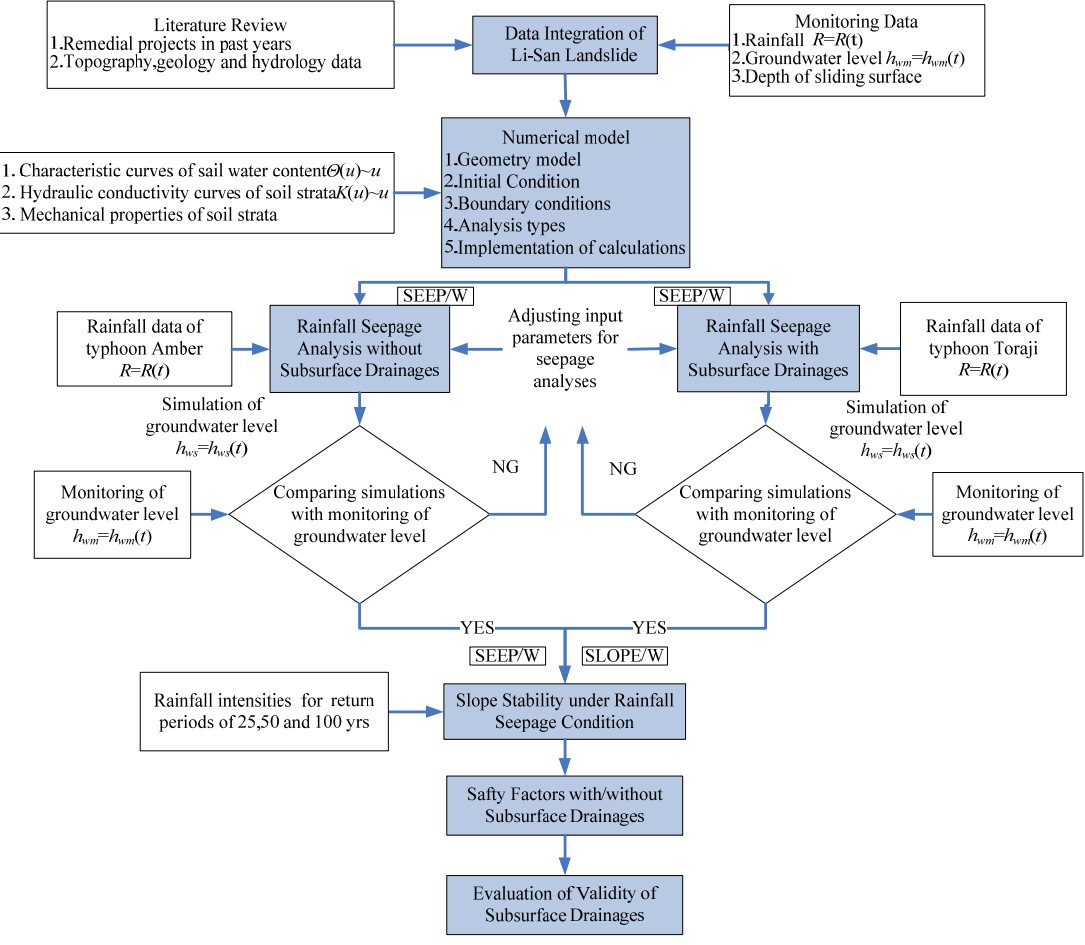

235   .
236            Fig. 11 Evaluation processes for the validity of subsurface drainage systyems in Li-Shan landslide

**3.2 Numerical Simulation of Subsurface Drainages**

240       The subsurface drainage systems in Li-Shan landslide is comprised of drainage wells and drainage galleries and their
drainage effects can be simulated by assigning a series of line-type and point-type drainage boundary conditions along the
drainage boreholes in the numerical model.
**(1) Drainage wells (Vertical shaft with drainage boreholes)**
244       It was assumed that the fan-shaped array of drainage boreholes is functional well without clogging during drainage. The
function of drainage boreholes installed at 3~4 different elevations in the vertical shaft (see Fig. 8 ) can be effectively simulated
by specifying a line-type free seepage surface boundary condition (potential free seepage face review $Q$=0) along the boreholes
as illustrated in Fig. 12. Through this free seepage face, the infiltrated rainwater above the surface was drained out of the
water-bearing layers. Nevertheless, it should be noted that it will be improper to assign a zero pressure head condition or
atmospheric condition (pressure head $h_p$=0) along the drainage borehole. If doing so, the portion of drainage borehole situates
above the groundwater level at unsaturated zone will possess a negative pressure head (for unsaturated zone, $h_p$<0) and
eventually extracts groundwater from saturated zone (for saturated zone, $h_p$>0) into unsaturated zone. However, this situation is
not the case in reality.
**(2) Drainage Galleries with Sub-vertical Drainage Boreholes**
254       An average 5 sub-vertical drainage boreholes with radial array along the crown arch of gallery per unit length of
water-collection chamber (out of plane) are fanning out into the water-bearing stratum to collect groundwater and which can be
simulated by assigning a point-type flow boundary on the 5 installation points of drainage boreholes, as the triangle points
illustrated in Fig. 12(a). In 2-D numerical model, the required input outflow rate of 5 point-type flow boundaries was estimated
according to the measurements of average outflow rate $Q_G$ ($Q_{G1}$= 60~90 $m^3$/hr, $Q_{G2}$=36~60 $m^3$/hr) of the two drainage galleries
$G1$ and $G2$. The drainage rate $q_G$ ($m^3$/hr-m) for each point-type drainage borehole unit length of water-collection chamber (out of
plane) can be estimated as:

$$q_G(\mathrm{m}^3/\mathrm{hr}\text{-}\mathrm{m}) = \left(\frac{Q_G}{l_G \times N_G}\right) \div N_C = \left(\frac{Q_G}{L_G}\right) \div N_C$$



In which, $N_G$=number of water-collection chamber along $G1$- and $G2$-gallery ($N_{G1}$=5, $N_{G2}$=10); $l_G$=length of water-collection
chamber along $G1$- and $G2$-gallery ($l_{G1}$= $l_{G2}$=6 m); $L_G$=total length of water-collection chamber along $G1$- and $G2$-gallery=$l_G \times$
$N_G$ ($L_{G1}$=6×5=30 m, $L_{G2}$=6×10=60 m). Moreover, $N_C$=number of radial drainage boreholes per unit length of water-collection
chamber=$N_{C1}$=$N_{C2}$=5. Eventually, using the above equation, one can insert 5 nodes (=$N_C$) with an assigned drainage boundary
condition of drainage rate $q_{G1}$= 0.5 m³/hr-m and $q_{G2}$= 0.16 m³/hr-m to each node for $G1$- and $G2$-gallery respectively.
**(3) Material Model Parameters.**
Prior to the typhoon rainfall event, the slide body above groundwater table comprised of colluviums and heavily to medium
weathered slate is unsaturated, the effects of matric suction (negative pore-water pressure) on the seepage and stability analyses
need to be considered. The hydraulic conductivity, $K(u_w)$, of slide body is not a constant whereas changes with the variation of
pore-water pressure, $u_w$. The soil water characteristic curve (or SWCC), $\Theta(u_w)\sim u_w$, defines the volumetric water content, $\Theta(u_w)$,
corresponding to a specific matric suction, $u_w$, and has significant effects on the hydraulic behaviors and shear strength of
unsaturated soil mass. The methods used to determine the SWCC have been studied by many researchers (Green and Corey,
1971; van Genuchten, 1980; Kovács, 1981; Arya and Paris, 1981; Fredlund and Xing, 1994; Aubertin et al., 2001) and most of
the methods are relevant to the grain size distribution curve and physical properties such as porosity and Atterberg′s limits of soil
sample. As a result, the SWCC is commonly applied to evaluate the hydraulic conductivity curve, $K(u_w)\sim u_w$, required for
seepage analysis. In this study, all the SWCC of soil strata are evaluated on the basis of grain size distribution curve. An
appropriate estimation of SWCC is very important for colluviums because it significantly affects the rainfall infiltration at the
onset of rainfall.

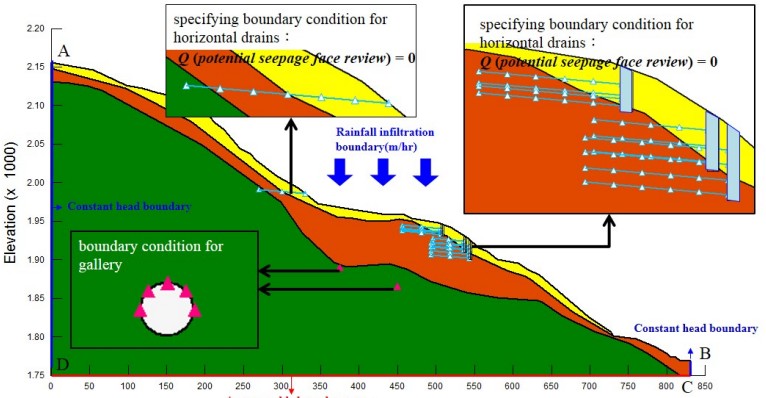

(a)

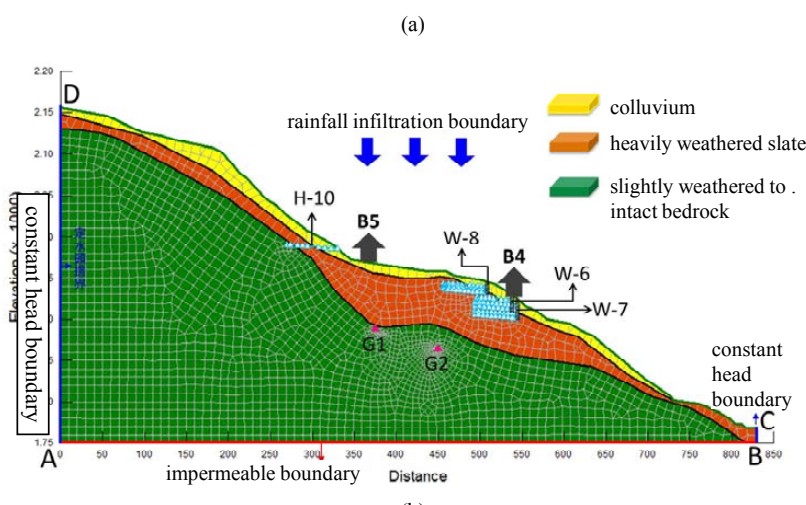

(b)

Fig. 12     boundary conditions for subsurface drainages (b) finite element mesh of geological cross section with draina_ wells of *W-6*,
*W-7*, *W-8*, and *H-10*; observation wells of *B4* and *B5* and drainage galleries *G1*, *G2* along *Y2-profile*

In this study, the $K(u_w)\sim u_w$ curve of soil stratum were determined according to the saturated hydraulic conductivity, $K_{sat}$,
obtained from field pumping test and the corresponding soil water characteristic curve (or SWCC), $\Theta(u_w)\sim u_w$, proposed by




Fredlund and Xing (1994). Tsaparas, et al. (2002) also pointed out that the $K_{sat}$ value has significant influence on the seepage
pattern within an unsaturated soil slope. The required input material model parameters for rainfall-induced seepage analyses and
the sequential slope stability analyses are summarized in Tables 1 and 2. Fredlund et al. (1996) developed a simple equation
based on the Mohr-Coulomb failure criterion to predict the shear strength of unsaturated soils. The $\phi^b$ angle in Table 2 is used to
consider the contribution of matric suction to the shear strength of unsaturated soil and approximates $10°\sim15°(=\phi^b=\phi'/2)$ for
practical purposes

Table 1 Input material model (unsaturated model) parameters for seepage analysis

| Soil Type | Saturated volumetric water content $\Theta_{sat}$ (%) | Saturated hydraulic conductivity $K_{sat}$ ($\times10^{-2}$ m/hr) |
|---|---|---|
| Colluviums | 0.281 | 5.868 |
| Heavily to medium weathered slate | 0.206 | 2.858 |
| Lightly weathered slate to intact bedrock | 0.230 | $4.9\times10^{-4}$ |

$\Theta=S\times n$ ; $\Theta_{sat}=1\times n=n$ ($S$ =degree of saturation, $n$ =porosity)


Table 2 Input material model (Mohr-Coulomb model) parameters for slope stability analysis

| Soil Type | Unit volumetric weight $\gamma$ (kN/m³) | Cohesion $c'$ (kPa) | Friction angle $\phi'$ (°) | Equivalent friction angle of matrix suction $\phi^b$ (°) |
|---|---|---|---|---|
| Colluviums | 17.07 | 10.79 | 27 | 10 |
| Heavily to medium weathered slate | 22.56 | 19.62 | 28 | 10 |
| Lightly weathered slate to intact bedrock | 27.06 | 294.3 | 33 | 0 |

(1) $\gamma$, $c'$ and $\phi$ are determined by field and laboratory tests.
(2) The modified Mohr-Coulomb failure criterion $\tau= [c'+(\sigma_n-u_a)\times\tan\phi'+(u_a-u_w)\times\tan\phi^b]$ is adopted for slope
  stability analysis. In which, $u_a$ and $u_w$ represent the pore-air and pore-water pressures of soil mass.
(3) In the above equation, the $\phi^b$ angle is used to consider the contribtion of matric suction to the shear
  strength of unsaturated soil.

**3.3 Implementation of Numerical Analyses**
Rainfall-induced seepage and slope stability analyses before and after subsurface drainages remediation was performed
along *Y2-profile* situates at the southeast region of Li-Shan landslide. Using SEEP/W (Geo-Studio, 2012) finite element method
(*FEM*) to calculate the groundwater levels variation and pore-water pressure distribution throughout the problem domain, which
is then introduced at the potential sliding surface at each time step into SLOPE/W (Geo-Studio, 2012 ) limit equilibrium method
(*LEM*) for the sequential slope stability analyses. Rainfall hyetographs of Typhoons Amber (1997) and Toraji (2001), as shown
in Fig.s 13 and 14, were used correspondingly for the analyses without and with remediation. The groundwater flow model is
then calibrated with groundwater levels variation measured from B5 monitoring station. It should be noted that the subsurface
drainage systems had not been completed during Amber Typhoon (1997/8/14~1997/8/28) while the meteorological condition
with large amounts of precipitation over a relatively short period during Toraji Typhoon (2001/7/29~2001/7/31) was extremely
adverse to the slope stability. In addition, Rahardjo (2001) indicated that the precedent rainfall has significant effects on slope
stability. An precedent rainfall with higher intensity and longer duration enables to preserve water content in soil mass and
expedite the infiltration of rainwater from the sequential torrential rainfall which eventually causes slope failure (Sitar, 1992;
Tsaparas et al., 2002). As a consequence, the precedent rainfalls of above two typhoon events were also considered in the
rainfall-induced transient seepage analyses of the landslide.
**(1) Rainfall-induced Seepage Analyses without Remediation.** Transient Seepage Analysis: (1) *First Stage*: the groundwater
level and pore-water pressure were calculated using 14 days precedent rainfall, as shown in Fig. 13(a), prior to Amber
Typhoon (1997). (2) *Second Stage*: feedback of groundwater level and pore-water pressure from (1) *First Stage* as initial
conditions, then the analysis was performed using the sequential rainfall of Amber Typhoon as shown in Fig. 13(b).
**(2) Rainfall-induced Seepage Analyses with Remediation.** Transient Seepage Analysis: (1) *First Stage*: the groundwater
level and pore-water pressure were calculated using 3 days precedent rainfall of Toraji Typhoon (2001). (2) *Second Stage*:
feedback of groundwater level and pore-water pressure from (1) *First Stage* as initial conditions, then the analysis was
performed using the sequential rainfall of Toraji Typhoon (2001) as shown in Fig. 14.
**(3) Slope Stability Analyses without and with Remediation.** Slope stability analysis (*LEM* analysis) was carried out using
the time-dependent pore-water pressure distribution $u_w(t)\sim t$ calculated from rainfall-induced seepage analysis (*FEM*
analysis). In *LEM* analysis, the Morgenstern-Price sliced method (Morgenstern and Price, 1965) which considered the
strict requirement of force equilibrium in derivations was adopted to calculate the time-dependent factor of safety $F_s=F_S(t)$
for 3 known *Potential Sliding Surfaces* (*1st-PSS*, *2nd-PSS* and *3rd-PSS*) for precedent rainfall duration $t=1\sim14$ day with $\Delta t=1$
day; and $t=1\sim41$ hour (Amber Typhoon, 1997) and $t=1\sim29$ hour (Toraji Typhoon, 2001) with $\Delta t=1$ hour, where $\Delta t=$time
increment as shown in Figs. 13 and 14.




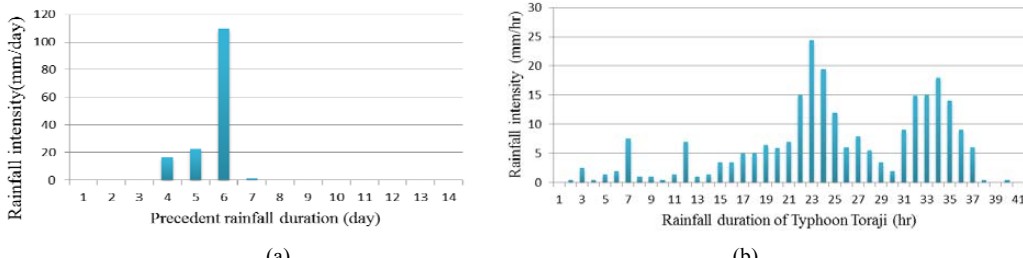

(a)                                                            (b)

Fig. 13 (a) 14 days precedent rainfall hyetograph prior to Amber Typhoon (1997/8/14~1997/8/28) (b) rainfall hyetograph of Amber Typhoon (1997/8/28~1997/8/29)

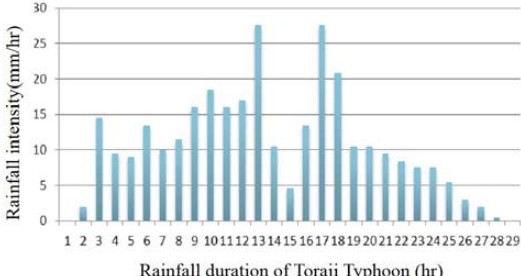

Fig. 14 Rainfall hyetograph of Toraji Typhoon (2001/7/29~2001/7/31)

## 4. Results and Discussions

### 4.1 Verification of Rainfall-Induced Seepage

As shown in Fig. 15, the groundwater level variation of simulation without (Fig. 15(a)) and with (Fig. 15(b)) subsurface drainages remediation is tiny and agree with those of observation from *B5* monitoring station. The slight variation of groundwater level at *B5* monitoring station could be resulted from the geological feature of thin colluviums and thick fractured slate underneath in this area because this makes difficult for the soil strata to accumulate the infiltrated rainwater from long duration rainfall and to raise the groundwater level. Conclusively, the proposed numerical procedures can properly simulate the groundwater level variation of *B5* monitoring station with and without subsurface drainages in Li-Shan landslide and the validities of numerical procedures and input model parameters were then verified. The numerical results of seepage analyses enable to provide more realistic and reliable pore-water pressure for the subsequent stability analyses.

### 4.2 Function of Subsurface Drainages

The objective of the Li-San landslide remediation using subsurface drainage systems aimed at reducing the peak piezometric heads in the slide body by 10~30 m (SWCB, 2003) and facilitate a quick drawdown of rising groundwater level during torrential rainfall. It can be found that the groundwater level (variation at a depth of 50~52 m in Fig. 15(b)) at *B5* monitoring station with subsurface drainages remediation during Toraji Typhoon (2001, peak rainfall intensity=27 mm/hr; rainfall duration $t$ =29 hrs) is about 40 m lower than that (variation at a depth of 10~10.5 m in Fig. 15(a)) without remediation during Amber Typhoon (1997, peak rainfall intensity=24 mm/hr; rainfall duration $t$=41 hrs). The large lowering of groundwater levels were mainly caused by the drainage wells (see *H-10*, *W-6*, *W-7*, and *W-8* in Fig. 12(b)) at the central area of the slope which can tap into the colluviums and weathered slate and effectively drain off the infiltrated rainwater in the slope. Meanwhile, to match the calculated groundwater level variations with the observed one, the hydraulic conductivity curve, $K(u_w)$~$u_w$, of various soil strata are made some adjustments and finally determined.

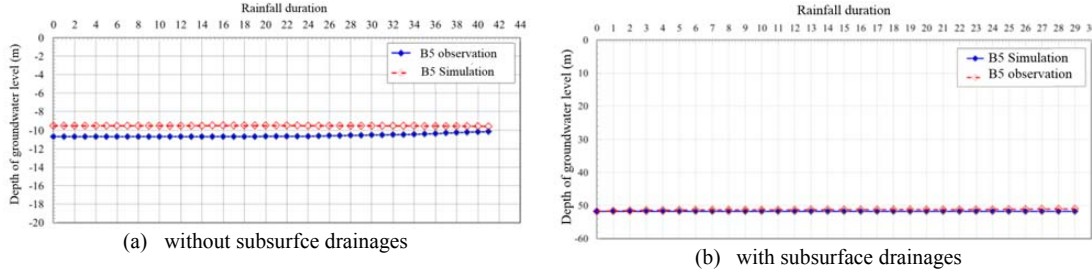

(a)  without subsurfce drainages                    (b)  with subsurface drainages

Fig. 15 Comparison of groundwater level variation between simulations and observations at *B5* monitoring station during rainfall of (a)Amber Typhoon (1997), and (b) Toraji Typhoon (2001)





Figure 16 illustrates the effect of the nearby drainage gallery (*G1*-gallery completed in 2001/01) on the long-term
(1997/01~2011/11) groundwater levels variation of *B5* monitoring station. The figure gives the elevations of *B5* borehole and
groundwater levels before/after construction of *G1*-gallery are 1,968, 1,945, and 1,917 m a.s.l. respectively. It was also indicated
that the groundwater levels were lowered down for about 28 m (=1,945 m-1,917 m) during five Typhoon events (Typhoons
Mindulle, Haitang, Longwang, Fungwong, and Morakot) after 2001/01.

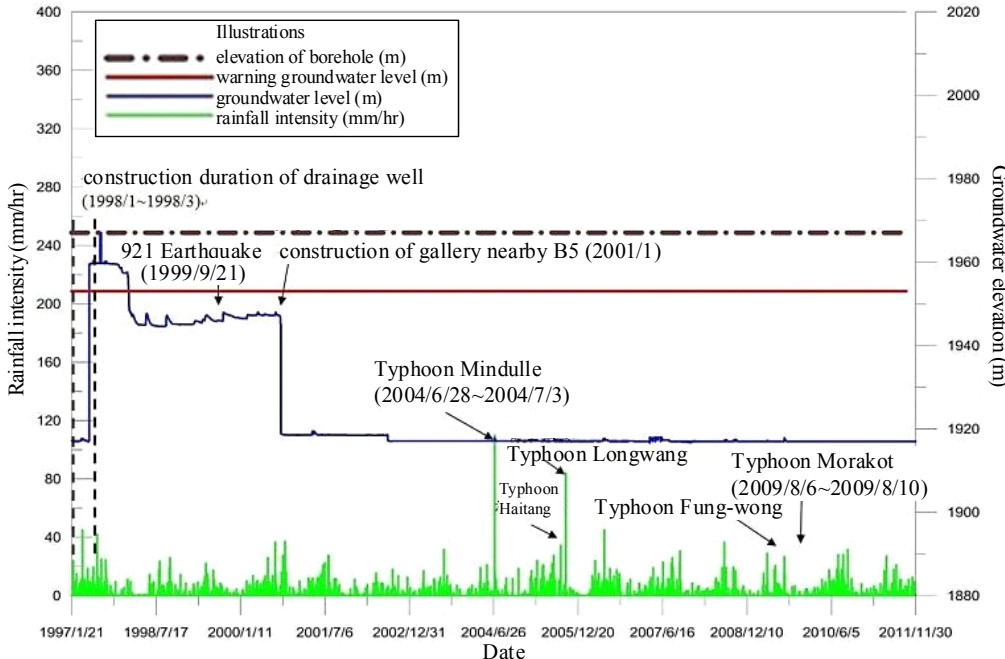

Fig. 16 Groundwater levels variation versus rainfall intensities of *B5* monitoring station (1997/01~2011/11)

**4.3  Stability of Potential Sliding Surfaces with and without Remediation**
The validity of subsurface drainages in Li-Shan landslide can be evaluated directly from the distribution of pore-water
pressure and the corresponding factor of safety, *Fs*, with and without remediation along *P*otential *S*liding *S*urface (*PSS*) or
indirectly from the distribution of volumetric water content within soil strata during rainfall. In cooperating the inclinometer
measurements with  stability analyses, three potential sliding surfaces, namely, *1st*-*PSS*, *2nd*-*PSS* and *3rd*-*PSS* as shown in Figs.
17 (a)~(c), can be determined along *Y2-profile* at southeast region of Li-Shan landslide. Their stabilities were diagnosed by
inspecting the pore-water pressure of monitoring points (*X1~X3* for *1st*-*PSS*; *Y1~Y3* for *2nd*-*PSS*; *Z1~Z3* for *3rd*-*PSS*) along
potential sliding surfaces. Generally, the *Fs* value of natural slope in the mountainous area of Taiwan is only slightly greater than
unity. Therefore, the slope tends to situate in a marginally stable state (*Fs*≈1.0) and is highly sensitive to heavy rainfall or
intensive earthquake. In Taiwan, three *Fs* values are adopted as technical criteria for slope engineering design: (1) for ordinary
time *Fs*≥1.50, (2) for earthquake *Fs*≥1.2, (3) for torrential rainfall *Fs*≥1.10. Popescu (2001) proposed a three-stage continuous
spectrum of *Fs* to define the stability state of slopes: *Fs*>1.3 (stable), 1.0< *Fs*<1.3 (marginally stable), and *Fs*<1.0 (actively
unstable). The factors of safety, *Fs*, of the three potential sliding surfaces with and without subsurface drainages were
summarized in Table 3. As listed in the table, a higher *Fs* value with lower decreasing percentage during rainfall is always
obtained for the case with subsurface drainages remediation (Toraji Typhoon, 2001) rather than the case without remediation
(Amber Typhoon, 1997).
Table 3 Factors of safety with and without subsurface drainages along potential sliding surfaces

| Potential Sliding Surface (PSS) | Factor of Safety $F_S$ | | | |
| | [1]without remediation | | [2]with remediation | |
| | Variation during rainfall | Decreasing percentage (%) | Variation during rainfall | Decreasing percentage (%) |
|---|---|---|---|---|
| *1st*-*Potential Sliding Surface* (*1st*-*PSS*) | 1.148→1.096 | 4.53 | 1.240→1.228 | 0.96 |
| *2nd*-*Potential Sliding Surface* (*2nd*-*PSS*) | 1.317→1.263 | 4.10 | 1.521→1.512 | 0.59 |
| *3rd*-*Potential Sliding Surface* (*3rd*-*PSS*) | 1.250→1.210 | 3.20 | 1.459→1.452 | 0.48 |

[1]Amber Typhoon in 1997 without remediation (Fig. 13) (rainfall duration *t*=41 hr), the subsurface
     drainages system has not been completed yet in this duration
[2]Toraji Typhoon in 2001 with remediation (Fig. 14) (rainfall duration *t*=29 hr)
[3] $F_S$ ≥1.1 for torrential rainfall; $F_S$ ≥1.5 for ordinary time (stability criteria used in Taiwan)



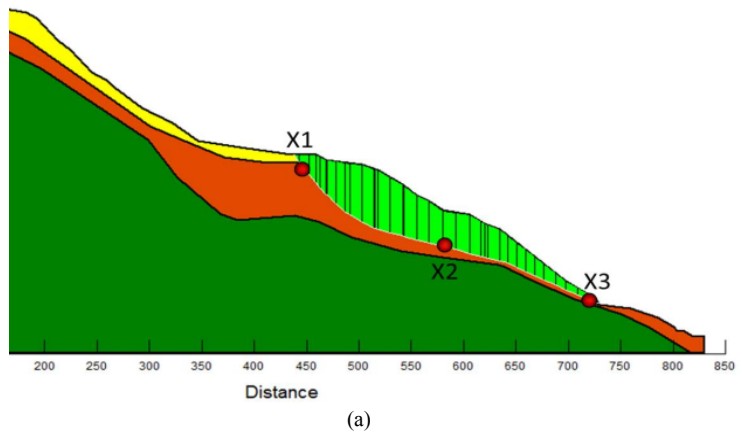


(a)

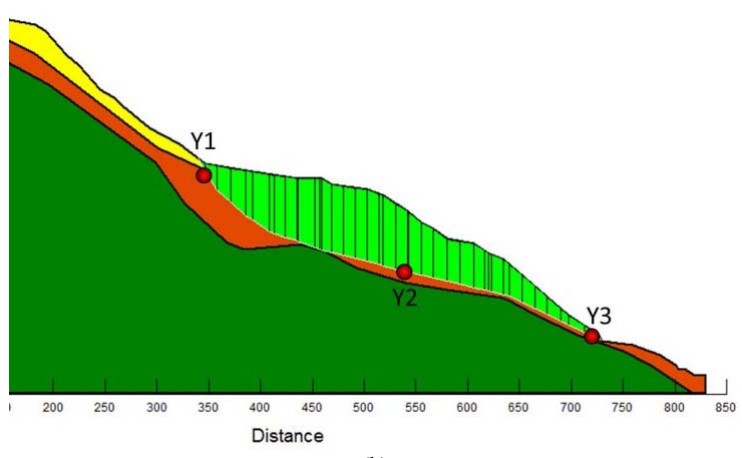


(b)

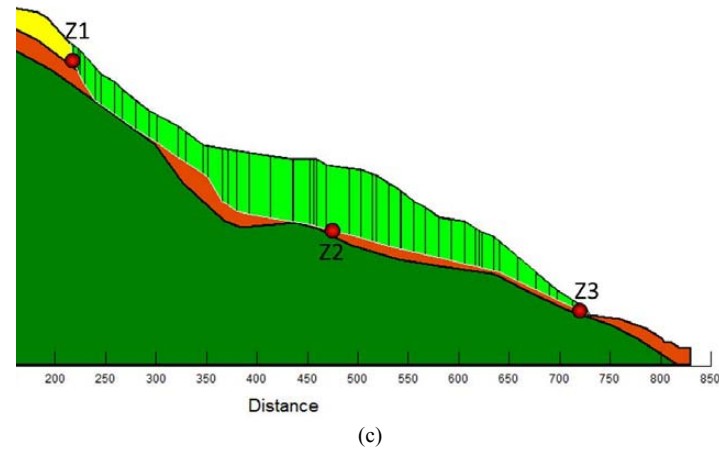


(c)

Fig. 17 three *Potential Sliding Surfaces* (*PSS*) along *Y2-profile* in Li-Shan landslide and their corresponding monitoring points (a) *X1~X3* for *1st-PSS*
(b)*Y1~Y3* for *2nd-PSS* (c) *Z1~Z3* for *3rd-PSS*


**(1) Pore-water pressure and Factor of Safety of *1st-PSS* during Two Typhoon Events**

According to the numerical results, the *Fs* value is greatly dependent on the relative locations between the potential sliding
surface and the groundwater level. In addition, the groundwater level is dominated by the interaction between rainfall infiltration
and subsurface drainage systems. Consequently, a higher factor of safety with lower decreasing rate during torrential rainfall for
a potential sliding surface is mainly attributed to the lower down of groundwater level and decrease of pore-water pressure
caused by subsurface drainage systems. Due to the similarity of numerical results for the three potential sliding surfaces, only
the factor of safety of *1st-PSS* (*1st-Potential Sliding Surface*, see Fig. 17(a)) $F_S$ with minimum value of 1.096 (see Table 3) and the



corresponding pore-water pressure of monitoring points *X1*, *X2* and *X3* were presented and discussed in detail. Two typhoon
events, Amber Typhoon (1997/8/28~1997/8/29; with 14-days precedent rainfall: 1997/8/14~1997/8/28) and Toraji Typhoon
(2001/7/29~2001/7/31) occurred at different durations were used for the numerical analyses of *Y2-profile* in Li-Shan landslide
for two situations, namely, without and with subsurface drainages remediation.
Comparing Fig. 12(b) with Fig. 17(a), it can be seen that the monitoring point *X2* of *1st-PSS* is immediately underneath the
drainage boreholes of vertical shafts *W-6*, *W-7* and *W-8* and in the vicinity of *G2*-gallery. In addition, the monitoring point *X1*
also situates at the down slope of drainage boreholes of vertical shaft *H-10*. These indicate the subsurface drainage systems have
crucial influence on the seepage behaviors of monitoring points *X1* and *X2* during rainfall. Further,  because of situating at a
lower elevation of slope toe, it is rational to evaluate the efficiency of subsurface drainages by inspecting the  response of
pore-water pressure of monitoring point *X3* which tends to accumulate the groundwater flows from upslope. The pore-water
pressure distribution of monitoring points *X1~X3* along *1st-PSS* is significantly dependent on the variation of groundwater level
calculated by the rainfall induced seepage analyses.
For the case without subsurface drainages remediation, as displayed in Fig. 18(a), before torrential rainfall, the initial
pore-water pressure ($u_w$ for rainfall duration $t$=0) of point *X1* ($u_w$=-261.4 kPa) and *X3* ($u_w$=-17.4 kPa) are negative (suction force)
due to situating above the groundwater level at unsaturated zone while point *X2* ($u_w$=124.6 kPa) is positive (squeeze force)
below the groundwater level. Comparing with the case with remediation, as shown in Fig. 18(b), the initial pore-water pressure
of points *X1~X3* are constantly lower than that without remediation (Fig. 18(a)) no matter the pressure is negative for points *X1*
($u_w$=-467.5 kPa) and *X3* ($u_w$=-22.3 kPa) or positive for point *X2* ($u_w$=92.8 kPa). This is attributed to the function of subsurface
drainages in the ordinary time of non-typhoon seasons.

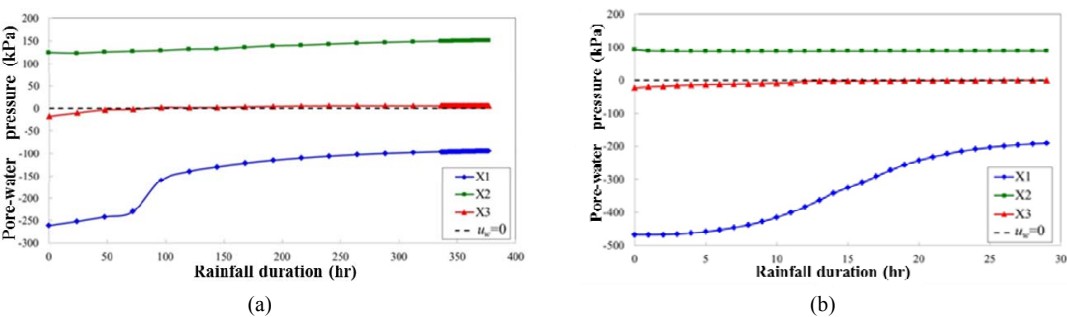

(a)                                                                          (b)

Fig. 18 Variation of pore-water pressure of *1st-PSS* (a) Amber Typhoon (1997) without remediation (b) Toraji Typhoon (2001) with remediation

During Amber Typhoon in 1997 (Fig. 18(a)), the subsurface drainages remediation has not functioned yet, the negative
pore-water pressure (or suction pressure) of point *X1* greatly decreases during rainfall ($u_w$=-261.4 kPa→-94.1 kPa) and the shear
strength of soil mass might alternately reduce because of soil matric suction loss. On the other hand, during Toraji Typhoon in
2001 (Fig. 18(b)), due to the function of subsurface drainages, although the suction loss ($u_w$=-467.5 kPa→-190.3 kPa) of point
*X*1 remains, the final suction pressure is still higher than that during Amber Typhoon ($u_w$=-190.3 kPa > $u_w$=-94.1 kPa). This
demonstrates the subsurface drainages enable to mitigate the softening and deterioration of wetting soil mass during torrential
rainfall and to prevent a rapid reduction of slope stability.
As shown in Fig. 18 (b), the positive pore-water pressure (or squeezing pressure) of point *X2* at the middle point of *1st-PSS*
(see Fig. 17(a)) with subsurface drainages remediation is lower than that without remediation (Figs. 18(a)) and situates in a
stable state throughout the entire rainfall duration under the function of subsurface drainages during Toraji Typhoon.
Additionally, comparing Fig. 18(a) and (b) for monitoring point *X2*,  the squeezing pressure of point *X2* increases gradually with
the rainfall duration ($u_w$=124.6 kPa→151.4 kPa) during Amber Typhoon in 1997 (Fig. 18(a)). On the contrary, the squeezing
pressure of point *X2* only appears slightly influenced by the infiltrated rainwater during Toraji Typhoon in  2001 (Fig. 18(b))
($u_w$=92.8 kPa→88.5 kPa) and eventually tends a steady condition. This implies the subsurface drainages can suppress an
increase of positive pore-water pressure and situate the slopes in a comparatively stable condition. According the numerical
reslutls, the stability of *1st-PSS* is influenced by deeper groundwater flow which cause pore-water pressure increasing on
potential sliding surface rather than by direct infiltration of ground surface. Similarly, Ng, CWW and Shi, Q (1998) pointed out
that rainfall leads to an increase in pore water pressure or a reduction in soil matric suction and in turn, results in a decrease in
shear strength on the potential sliding surface.
As shown in Figs. 18(a) and (b), the groundwater flow eventually tends to accumulate at the monitorning point *X3*, due to
the point situating at the lower elevation of *1st-PSS* with very thin colluviums overburden (see Fig. 17(a)), the minor suction of
point *X3* decreases gradually into a lower level of nearly zero value ($u_w$=-17.4 kPa→0 kPa, for Amber in 1997; $u_w$=-22.3 kPa →0
kPa, for Toraji in 2001) during the rainfalls of the two typhoons. The subsurface drainages remediation has little effect on the
point *X3* where is in vicinity of the outlet of the potential sliding surface.
In conclusion, the cumulative groundwater in the heavily to medium weathered slate above the *1st-PSS* and the rainwater
perched between the colluviums and heavily to medium weathered slate was drained out of the sliding mass through drainage
galleries *G1* and *G2* in a short period. It should be noted that the drainage galleries always situate at the intact fresh slate and
underneath the potential sliding surface (see Fig. 12(b)). Finally, the pore-water pressure distributions in Fig. 18 were then used
to calculate the corresponding factor of safety $F_S$ values of *1st-PSS* during typhoons, as shown in Fig. 19. For the case without
subsurface drainages remediation (Fig. 19(a)), the $F_S$ values are descending with elapsed time to a minimum value of 1.096
(=$F_{Smin}$) during Amber Typhoon. Comparatively, for the case with remediation (Fig. 19(b)), the $F_S$ values are constantly higher



Natural Hazards
and Earth System
than those of without remediation and come to a minimum value of 1.228 ($=F_{Smin}$) and almost not affected by Toraji Typhoon.
This demonstrates that the subsurface drainage systems can function effectively to intercept the groundwater flow from
infiltrated rainwater and largely mitigate the rising potential of pore-water pressure on the potential sliding surface which
alternately enables to maintain the slope in a certain stability level during rainfall.

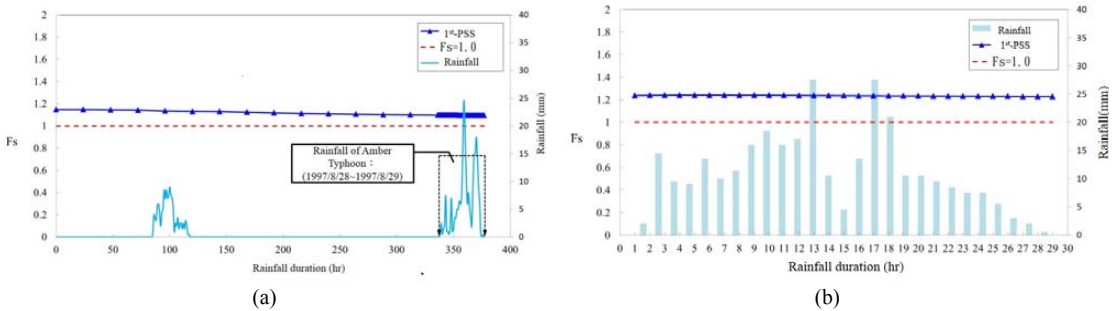

(a)                                                     (b)

Fig. 19 Variation of factor of safety of $1^{st}$-$PSS$ (a) Amber Typhoon (1997/8/28~1997/8/29) without remediation (b) for Toraji Typhoon
(2001/7/29~2001/7/31) with remediation

**(2) Effects of Fictitious Subsurface Drainages on the Slope Stability of $2^{nd}$-$PSS$ during Amber Typhoon**
To understand the effect of subsurface drainages on the slope stability of landslide, a numerical experiments were carried
out using Amber Typhoon (1997/8/28~1997/8/29; with 14-days precedent rainfall: 1997/8/14~1997/8/28; see Fig. 13) for the
seepage and slope stability analyses of $2^{nd}$-$PSS$ along the *Y2-profile* of Li-Shan landslide with (fictitious) and without subsurface
drainages remediation. Due to the fact that the remediation had not been completed yet during Amber Typhoon in 1997, the
drainage wells and drainage galleries were assumed fictitiously to be functional and simulated by assigning specific flow
boundary conditions in numerical model. The factors of safety, $Fs$, of the three potential sliding surfaces with and without
subsurface drainages were summarized in Table 4.
Table 4 Factors of safety without and with fictitious subsurface drainages along potential sliding surfaces during Amber Typhoon in 1997

| | Factor of Safety $F_S$ | | | |
| Potential Sliding Surface (PSS) | [1]without remediation | | [2]with fictitious remediation | |
| | Variation during rainfall | Decreasing percentage (%) | Variation during rainfall | Increasing percentage (%) |
|---|---|---|---|---|
| $1^{st}$-Potential Sliding Surface ($1^{st}$-PSS) | 1.148→1.096 | 4.53 | 1.149→1.201 | 4.53 |
| $2^{nd}$-Potential Sliding Surface (**$2^{nd}$-PSS**) | **1.317→1.263** | **4.10** | **1.351→1.403** | **3.85** |
| $3^{rd}$-Potential Sliding Surface ($3^{rd}$-PSS) | 1.250→1.210 | 3.20 | 1.304→1.409 | 8.05 |

[1]Amber Typhoon in 1997 (Fig. 13) (rainfall duration $t$=377 hr) without remediation, the subsurface
drainages had not been completed yet in this duration.
[2]Amber Typhoon in 1997 (Fig. 13) (rainfall duration $t$=377 hr) with remediation, the subsurface
drainages was fictitiously assigned in numerical model.
[3] $F_S$ ≥1.1 for torrential rainfall; $F_S$ ≥1.5 for ordinary time (stability criteria used in Taiwan)

Figure 20 shows that during Amber Typhoon the $F_S$ value of $2^{nd}$-$PSS$ with subsurface drainages ($F_S$=1.403 at the end of
rainfall, for $t$=377 hr) is constantly higher than those without drainages ($F_S$=1.263 at the end of rainfall, for $t$=377 hr) and the
potential effect of subsurface drainage systems is evaluated in term of the promotion percentage of $F_S$ value approximates 11.1%
(=[1.403-1.263]×100%/[1.263]). This demonstrates the subsurface drainage systems are effective on promoting the slope
stability of landslide. Meanwhile, as shown in Fig. 21, prior to the torrential rainfall, the potential sliding surface was submerged
by initial groundwater level and subsequently at the elapsed time of typhoon rainfall, $t$=23 hr, for the occurrence of peak rainfall
intensity, the groundwater level ascends for the case without drainages (Fig. 21(a)) and leads to a factor of safety $F_S$=1.264. On
the contrary, it becomes obvious that a groundwater drawdown for the case with subsurface drainages (Fig. 21(b)) and a higher
factor of safety $F_S$=1.399 can be achieved. The promotion percentage of $F_S$ value is about 10.7% for a rainfall duration of $t$=23 hr.
These results coincide with the study performed by Rahardjo and Leong (2002) that the horizontal drains (or drainage boreholes)
are mainly effective to improve the stability of the slope by lowering the groundwater table. Based on the numerical analyses of
a field instrumentation case, Rahardjo et al. (2012) also indicated that the $F_S$ values for the slope without horizontal drains are
much lower than those of the slope with horizontal drains. Santoso et al. (2009) investigated the influence of (length/spacing)
ratio of horizontal drains on residual soil slope stability and found that the promotion percentage of $F_S$ value approximates
12~15 % for a (length/spacing) ratio ranges from 4~9.





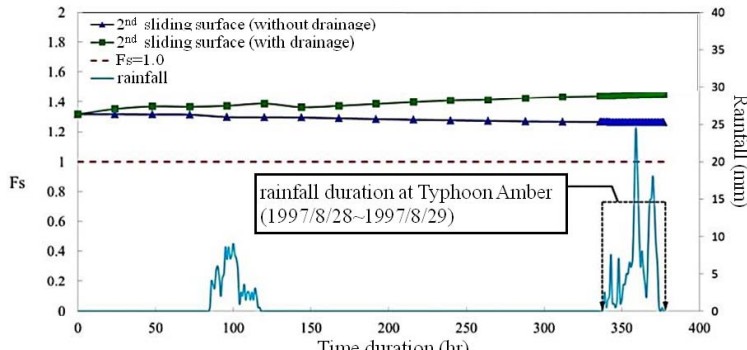

Fig. 20 The factors of safety of $2^{nd}$-PSS during Amber Typhoon
(1997/8/28~1997/8/29; with 14-days precedent rainfall during 1997/8/14~1997/8/28)

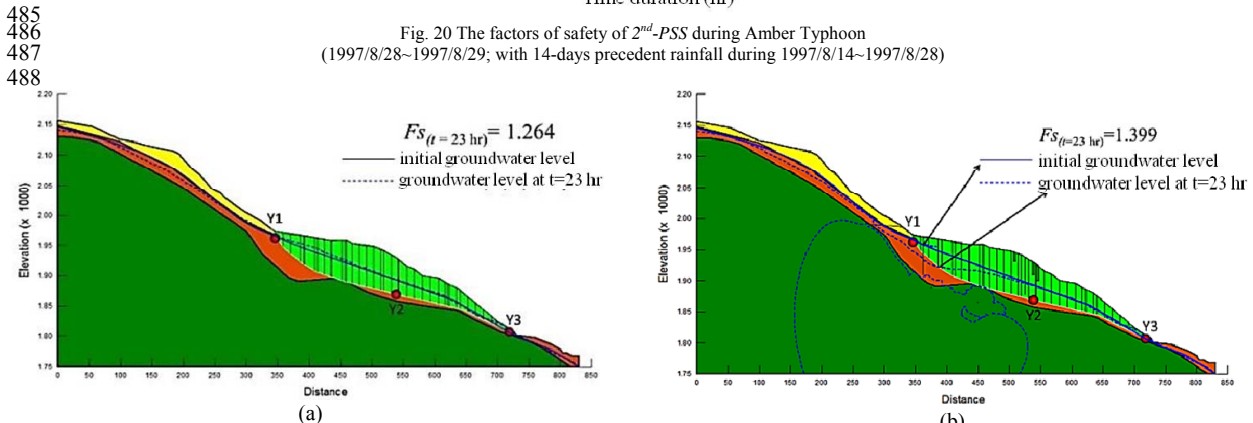

(a)   (b)

Fig. 21 Groundwater levels and factors of safety of $2^{nd}$-PSS at the rainfall duration $t$=23 hr
during Amber Typhoo (1997/8/28~1997/8/29) (a) without subsurface drainages (b) with fictitious subsurface drainages

Greco et al. (2010) indicated that monitoring of soil volumetric water content seemed more useful than soil suction monitoring for early warning purposes, since water content grew smoothly during the entire infiltration processes, while soil suction showed abrupt steep fronts. As illustrated in Fig. 22, the volumetric water contents $\Theta$ (=$S \times n$=0.05~0.20, in which, $S$ =degree of saturation, $n$ =porosity) of colluviums and heavily to medium weathered slate around the drainage galleries $G1$ and $G2$ during Amber Typhoon are lower than their saturated volumetric water content $\Theta_{sat}$ ($\Theta_{sat}$ =$S \times n$=1×0.281=0.281 for colluviums and $\Theta_{sat}$=$S \times n$=1×0.206=0.206 for heavily to medium weathered slate). These reveal that in addition to contributions to groundwater drawdown and pore-water pressure mitigation, the drainage galleries enable to convert the surrounding soil strata from submerged saturation into unsaturated condition ($\Theta < \Theta_{sat}$) which in turn improve the shear strength of soil mass and the stability of slope.

**(3) Volumetric Water Content during Two Typhoon Events**

Figure 23 illustrates the variation of volumetric water content $\Theta$ of soil strata with depth at $B4$ monitoring station without and with subsurface drainages remediation. For the case without remediation (Fig. 23(a)) during Amber Typhoon (1997/8/28~1997/8/29), the $\Theta$ values are descending gradually to a depth of -30 m under unsaturated condition when comparing with the saturated volumetric water content $\Theta_{sat}$ ($\Theta < \Theta_{sat}$). For colluviums in a depth of 0~-16 m and heavily to medium weathered slate of -16~-30 m, their $\Theta_{sat}$ values are equivalent to 0.281 and 0.206 respectively. On the contrary, for a depth ranges from -30 to -50 m, the $\Theta$ values start to ascend due to approaching the groundwater level which situates at a depth of around -50 m. Eventually for a depth larger than -50 m, the soil strata are completely submerged and saturated below groundwater level ($\Theta = \Theta_{sat}$=0.206).

On the other hand, for the case with remediation (Fig. 23(b)) during Toraji Typhoon (2001/7/29~2001/7/31), the volumetric water content $\Theta$ of colluviums near ground surface increases with the rainfall duration from 0.188 ($t$=5 hr) to 0.225 ($t$=29 hr) due to rainwater infiltration and the $\Theta$ value for a depth of 0~-10 m resembles to the tendency of the case without remediation. Subsequently, for a depth of -10~-20 m, although the soil stratum changes from colluvium to heavily to medium weathered slate at -16 m depth, the $\Theta$ values are decreasing with depth constantly from -10 to -20 m to a minimum value of $\Theta$=0.03. However, the volumetric water content $\Theta$ of soil strata adjacent to the ground surface for a depth of 0~-20 m never go beyond the saturated volumetric water content $\Theta_{sat}$ ($\Theta < \Theta_{sat}$=0.281).

It should be noted that $B4$ monitoring station is in the vicinity of drainage wells $W$-6, $W$-7 and $W$-8 (see Fig. 12(b)). At three different elevation levels from -20 to -40 m along the drainage wells, a series of drainage boreholes were drilled upward into the upslope of sliding body to collect groundwater, consequently the lower volumetric water content of soil strata within this depth range is expectable. Similarly, the $\Theta$ values start to increase from the depth of -40 to -60 m due to closing groundwater level and which locates at a depth of around -60 m ($\Theta_{sat}$=0.206) lower than -50 m for the case without drainage remediation (Fig. 23(a)). This also verifies that the drainage boreholes are of great advantage to the groundwater drawdown during torrential rainfall.




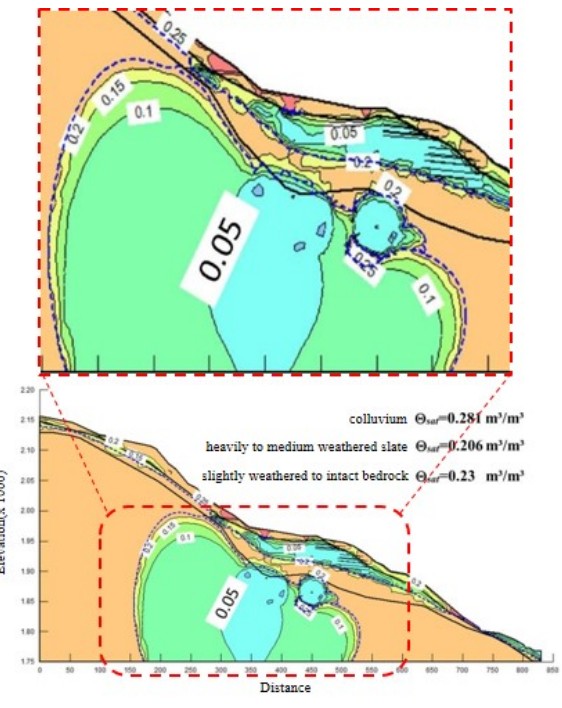

Fig. 22 Contour distribution of volumetric water content of soil strata surrounding drainage galleries *G1* and *G2*

**(4) Volumetric Water Content during Two Typhoon Events**
Figure 23 illustrates the variation of volumetric water content $\Theta$ of soil strata with depth at *B4* monitoring station without
and with subsurface drainages remediation. For the case without remediation (Fig. 23(a)) during Amber Typhoon
(1997/8/28~1997/8/29), the $\Theta$ values are descending gradually to a depth of -30 m under unsaturated condition when comparing
with the saturated volumetric water content $\Theta_{sat}$ ($\Theta<\Theta_{sat}$). For colluviums in a depth of 0~-16 m and heavily to medium
weathered slate of -16~-30 m, their $\Theta_{sat}$ values are equivalent to 0.281 and 0.206 respectively. On the contrary, for a depth ranges
from -30 to -50 m, the $\Theta$ values start to ascend due to approaching the groundwater level which situates at a depth of around -50
m. Eventually for a depth larger than -50 m, the soil strata are completely submerged and saturated below groundwater level
($\Theta=\Theta_{sat}$=0.206).
On the other hand, for the case with remediation (Fig. 23(b)) during Toraji Typhoon (2001/7/29~2001/7/31), the
volumetric water content $\Theta$ of colluviums near ground surface increases with the rainfall duration from 0.188 (*t*=5 hr) to 0.225
(*t*=29 hr) due to rainwater infiltration and the $\Theta$ value for a depth of 0~-10 m resembles to the tendency of the case without
remediation. Subsequently, for a depth of -10~-20 m, although the soil stratum changes from colluvium to heavily to medium
weathered slate at -16 m depth, the $\Theta$ values are decreasing with depth constantly from -10 to -20 m to a minimum value of
$\Theta$=0.03. However, the volumetric water content $\Theta$ of soil strata adjacent to the ground surface for a depth of 0~-20 m never go
beyond the saturated volumetric water content $\Theta_{sat}$ ($\Theta < \Theta_{sat}$=0.281).
It should be noted that *B4* monitoring station is in the vicinity of drainage wells *W-6*, *W-7* and *W-8* (see Fig. 12(b)). At three
different elevation levels from -20 to -40 m along the drainage wells, a series of drainage boreholes were drilled upward into the
upslope of sliding body to collect groundwater, consequently the lower volumetric water content of soil strata within this depth
range is expectable. Similarly, the $\Theta$ values start to increase from the depth of -40 to -60 m due to closing groundwater level and
which locates at a depth of around -60 m ($\Theta_{sat}$=0.206) lower than -50 m for the case without drainage remediation (Fig. 23(a)).
This also verifies that the drainage boreholes are of great advantage to the groundwater drawdown during torrential rainfall.
Slope stability analyses have indicated that rainwater infiltration results in a change of suction force and pore-water
pressure and the variation of groundwater level is the primary factor affecting the stability of slide mass in Li-San landslide. The
factor of safety against failure on the three potential sliding surfaces in Y2-profile that passing below the phreatic surface can be
improved by subsurface drainages. The increase of unit weight and decrease of shear strength that experienced by the
colluviums during torrential rainfall cause the southeast region of Li-Shan landslide particularly susceptible to instability. The
subsurface drainages remediation in Li-Shan landslide appears to have been very successful in attaining its objectives and the
groundwater levels monitoring data reported have met the requirements of drawdown. Only minor creep movements were
measured from field instrumentation in the past years.



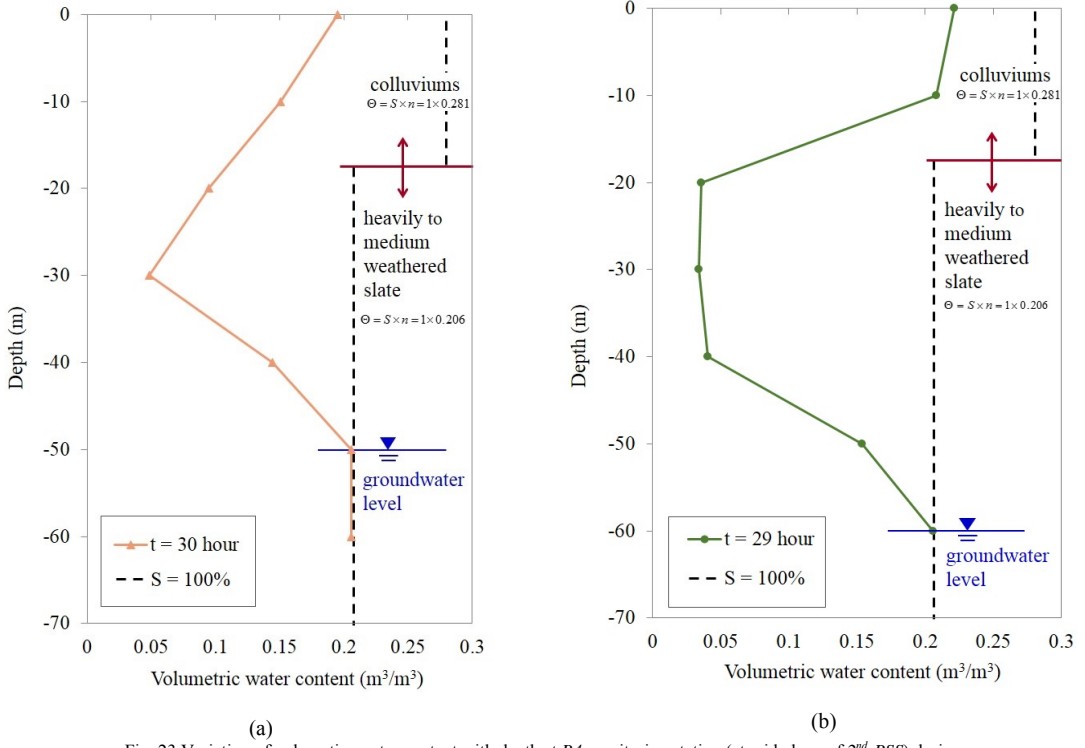

(a)                                    (b)

Fig. 23 Variation of volumetirc water content with depth at *B4* monitoring station (at mid-slope of *2nd-PSS*) during
(a) Amber Typhoon (1997) without remediation (b) Toraji Typhoon (2001) with remediation

**(5)  Effect of Rainfall Intensity with Different Return Period on Slope Stability**
To investigate the effect of rainfall intensity on the stability of Li-Shan landslide and validity of subsurface drainages, three
48-hr design rainfalls with retrun period of 25, 50 and 100 years for central Taiwan were used for rainfall induced seepage and
stability analyses of the three potential sliding surface in *Y2-profile* with subsurface drainages remediation. Incorporating the
rainfall distribution percentage of central Taiwan into rainfall frequency analyses, the design rainfalls can be obtained as shown
in Figs. 24(a)~(c).

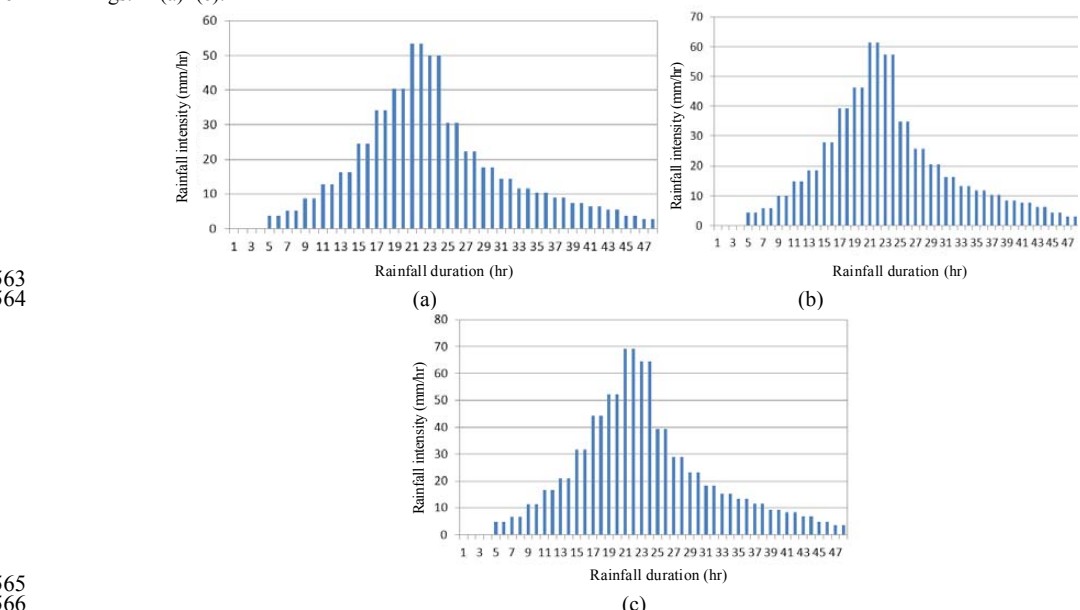

(a)                                                          (b)

(c)
Fig. 24 48-hr design rainfall with return period of (a) 25 years (b) 50 years (c) 100 years for central Taiwan





According to the numerical results, the three design rainfalls with return period of 25, 50 and 100 years have only minor
effect on the factors of safety of the three potential sliding surfaces (see Fig. 17) as shown in Table 5. The factors of safety $F_S$
corresponding to the three potential sliding surfaces ($1^{st}$-PSS, $2^{nd}$-PSS and $3^{rd}$-PSS) only decrease slightly
($F_S$=1.222→1.220→1.217 for $1^{st}$-PSS) in response to the three design rainfalls. Meanwhile, the $F_S$ values also constantly
maintain higher than unity ($F_S$>1.0 or $F_S$ ≥1.217) in the entire rainfall duration ($t$=48 hr). As a result, it can be deduced that the
capacity of subsurface drainage systems in Li-Shan landslide is sufficient to expedite the drainage of infiltrated rainwater
induced from high intensity and long duration rainfall and eventually to maintain the slope stability at a certain standard without
further deterioration.
Table 5 Factors of safety of three potential sliding surfaces for 48 hr rainfall duration under design rainfalls with different return periods

| Potential Sliding Surface (PSS) | Factor of Safety $F_S$ | | |
|---|---|---|---|
| | Return period of 25 years | Return period of 50 years | Return period of 100 years |
| $1^{st}$-Potential Sliding Surface ($1^{st}$-PSS) | 1.222 | 1.220 | 1.217 |
| $2^{nd}$-Potential Sliding Surface ($2^{nd}$-PSS) | 1.507 | 1.505 | 1.502 |
| $3^{rd}$-Potential Sliding Surface ($3^{rd}$-PSS) | 1.453 | 1.452 | 1.450 |

$F_S$ ≥1.1 for torrential rainfall; $F_S$ ≥1.5 for ordinary time (Slope stability criteria in Taiwan)

**5.   Conclusions**
The proposed numerical model is capable of capturing the groundwater responses of sliding body along the *Y2-profile* at
the southeast region of Li-Shan landslide during Amber (1997) and Toraji (2001) Typhoons. In numerical model, the functions
of subsurface drainages can be successfully modeled by assigning a line-type free seepage boundary along drainage boreholes
for drainage wells and a point-type flow boundary on drainage boreholes for drainage galleries. For Li-Shan landslide, the
factors of safety of the three potential sliding surfaces are nearly not influenced by torrential rainfall during Toraji Typhoon after
subsurface drainages remediation. Numerically, the subsurface drainages can expedite the drainage of infiltrated rainwater and
drawdown of groundwater level to maintain the slope stability at an acceptable standard during torrential rainfall. In addition,
the functions of subsurface drainage systems can be verified through the descending volumetric water content of soil strata
surrounding the drainage galleries or in a depth from -20 to -40 m of *B4* monitoring station where three levels of drainage
boreholes (or horizontal drains) were drilled for groundwater drainage. In addition, as the return period of design rainfall
increasing from 25 years to 100 years, although the factor safety of potential sliding surfaces $F_S$ exhibit a slight decreasing trend
for the entire rainfall duration, the $F_S$ values remain constantly greater than unity ($F_S$>1.0). As a consequence, the subsurface
drainage systems of Li-Shan landslide can function well to cope with the infiltration rainwater resulted from torrential rainfall
with high intensity and long duration and to prevent the slope from further deterioration. To date, no significant ground
movement of the landslide was instrumented after the completion of the subsurface drainage systems.

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
