# Peer review of "© Author(s) 2016. CC-BY 3.0 License."

_Natural Hazards and Earth System Sciences, 2015_

## Referee Comment (RC1) · Anonymous Referee #1 · 5 Feb 2016

It is a work in the form of presentation of a specific application. The manuscript is very complicated and understandable. The paper lacks of a coherent structure, a proper introduction, a brief review of the state of the art, and most importantly, it does not provide any original contribution to landslide studies. In the study, It has been showed that Factor of Safety increases the stability of the drainage work. This is the natural result of the landslide study. Therefore, scientific contribution of the study is very low.

---

## Referee Comment (RC2) · Anonymous Referee #2 · 11 Feb 2016

General Comments:

An interesting case study that highlights the role of drainage systems to reduce pressure heads within a landslide area is presented in the paper. The manuscript is clear and the scientific quality is generally good, although some parts lose strictness and need to be strengthened and improved. The structure of the paper can be improved, since a literature review on the specific topic (stabilization of large landslides with deep drainage interventions) lacks in the current version of the manuscript and must be added. Probably, the initial part of the manuscript (that is the general description of the landslide and the drainage works; sections 1 and 2) is too detailed and can be shortened, thus leaving space to the state-of-the-art discussion. In some parts the text is prolix, while the authors should better highlight the specific contribution of their scientific work.

[Figure]

Specific comments:

- A specific section on the subsurface hydraulics of the slope before intervention (from where the authors have started in their study) is needed, so that pre-intervention field measurements, monitoring stations, soil hydraulic parameters, along with the results of pre-intervention seepage analyses should be reported in detail. This part could be really helpful to clarify the role of drainage in the slope hydraulics.

- The authors should better clarify the choice of assigning in their seepage analysis a "free seepage surface boundary condition" to the horizontal drainages, instead of a zero pressure head boundary condition. The current explanation of the choice is unclear.

- How were the soil strength parameters reported in Table 2 chosen? This point requires a discussion from the authors.

- Vertical scale in Figure 15 is not adequate. The authors should enlarge the scale of y-axis, so that the trend can be better appreciated. In particular, it seems that in Fig. 15a the simulation is not capable of catching the effect of cumulated rainfall, as observed in reality.

- Table 4: it is unclear why Fs values increase during the rainfall history. This is not possible, since the effect of drainage should be a lower reduction of Fs respect to the pre-intervention situation, but not an increment. Fs could eventually increases in the long-term due to the effect of a drainage, but not during a rainfall event. A comment from the authors on this point is required. Also, vertical scale in Fig. 20 should be enlarged to appreciate the trends.

- Related to the previous point, it is unclear why groundwater level in Fig. 21 lowers as respect to the initial groundwater level during a rainfall history.

Technical comment:

- section 3 is completely repeated in section 4. Please remove section 3.

---

## Author Comment (AC1) · 28 Mar 2016

To Handling Editor: Paola Reichenbach

Dear Editor,

The co-authors of MS#nhess-2015-309, entiled Evaluating the Efficiency of Subsurface Drainages for Li-Shan Landslide in Taiwan, are still working on the revised process. Considering the revising quality, could you be so kind to extend the time of the revise process? Thank you for your assistance in this matter.

Sincerely,

Chan

H. C. Chan, Associate professor Department of Soil and Water Conservation National

Chung Hsing University 250 Kuo Kuang Rd., Taichung 402, Taiwan R.O.C. Tel: 886-4-22840381 ext 118 Fax: 886-4-22876851

---

## Author Comment (AC2) · 22 Apr 2016

It is a work in the form of presentation of a specific application. The manuscript is very complicated and understandable. The paper lacks of a coherent structure, a proper introduction, a brief review of the state of the art, and most importantly, it does not provide any original contribution to landslide studies. In the study, it has been showed that Factor of Safety increases the stability of the drainage work. This is the natural result of the landslide study. Therefore, scientific contribution of the study is very low. Responses: (1) The coherent structure, proper introduction and brief review of the state-of-the-art have been strengthened and improved. Please see the revised manuscript. (2) The paper provides a possible computation method and a quantitative indication to make a quick evaluation on the validity and efficiency of a subsurface drainage system with relatively high engineering costs. (3) The study is not designed for a scientific derivation or formulation of equations but for providing a practical example to those engineers who are working on the engineering construction and numerical analyses of subsurface drainage system in a large scale of landslide.

Please also note the supplement to this comment:
http://www.nat-hazards-earth-syst-sci-discuss.net/nhess-2015-309/nhess-2015-309-AC2-supplement.pdf

[Figure]

**Supplement:**

**Evaluating the Efficiency of Subsurface Drainages for Li-Shan Landslide in Taiwan**

Der-Guey Lin[1], Sheng-Hsiung Hung[2], Cheng-Yu Ku[3], Hsun-Chuan Chan[4]*

[1]Professor, Department of Soil and Water Conservation, National Chung-Hsing University

[2]Doctoral student, Department of Soil and Water Conservation, National Chung-Hsing University

[3]Professor, Department of Harbor and River Engineering, National Taiwan Ocean University

[4]Associate Professor, Department of Soil and Water Conservation, National Chung-Hsing University

[4]hcchan@nchu.edu.tw (*corresponding authorsd: No. 250 Kuo-Kuang Road, Taichung 402, Taiwan)

**Abstract:** This study investigates the efficiency of subsurface drainage systems includes drainage wells (vertical shaft with drainage boreholes or horizontal drains) and drainage galleries (longitudinal tunnel with sub-vertical drainage boreholes) for the slope stabilization of Li-Shan landslide in central Taiwan. The efficiency of the subsurface drainages is verified through a series of two-dimensional (2-D) rainfall induced seepage and slope stability analyses without and with subsurface drainages remediation during two typhoon events. Numerical results and monitoring data both show that the groundwater level at *B5* monitoring station with subsurface drainages remediation during Toraji Typhoon (2001) is about 40 m lower than that without remediation during Amber Typhoon (1997), and the factor of safety *Fs* of the first potential sliding surface (*1st-PSS*, the most critical potential sliding surface) is promoted simultaneously from 1.096 to 1.228 due to the function of subsurface drainage systems. In addition, the *Fs* values of the three potential sliding surfaces (*1st-PSS*, *2nd-PSS*, and *3rd-PSS*) stabilized by subsurface drainage systems are constantly maintained greater than unity ($F_S>1.0$ or $F_S \geq 1.217$) during rainfalls with return periods increases from 25 to 50 and 100 years. This demonstrates the subsurface drainage systems in Li-Shan landslide are functional and capable of accelerating the drainage of infiltration rainwater induced from high intensity and long duration rainfall and protect the slope of landslide from further deterioration. This study provides a quick computation method to evaluate the effectiveness and efficiency of a subsurface drainage system with relatively high engineering costs for a large landslide.

**Keywords:** landslides, subsurface drainage systems, drainage boreholes, drainage well, drainage gallery, potential sliding surface, factor of safety

**1. Introduction**

The Li-Shan landslide has a long history of intermittent large movements toward down slope during rainfall since 1990 and currently has been stabilized by the subsurface drainage systems implemented by Taiwan government for seven years′ duration from 1995 to 2002. The main remediation work for Li-Shan landslide was to lower the groundwater level through different subsurface drainage systems consisted of drain wells and drainage galleries. For subsurface drainage system, the effectiveness of horizontal drains on lowering the groundwater level and improving the slope stability is mainly governed by drain length, spacing, number, and installation location (Kenney et al., 1977; Prellwitz, 1978; Nonveiller, 1981; Lau and Kenney, 1984; Nakamura, 1988).

The function of horizontal drains has also been investigated by several numerical analyses. Cai et al. (1998) concluded that lengthening the horizontal drains is more effective than shortening the spacing and increasing the number of the horizontal drains in a group. Rahardjo et al. (2002, 2003, 2012) found that horizontal drains have little role in minimizing infiltration in an unsaturated residual soil slope and benefits of using horizontal drains can be obtained through the lowering of the groundwater table. In addition, the general trends of numerical results indicate the most benefit was derived from the drain located at the bottom of the slope. However, in the above analyses, the depth of horizontal drains is in comparatively shallow and unlike the deep drainage of Li-Shan landslide achieved by a series of drainage wells and drainage galleries. In addition, numerical modeling results, based on hydro-mechanically coupled 2-D distinct-element models, strongly suggest that deep drainage (drainage tunnel) was the key measure to successfully stabilize large landslides (Eberhardt et al., 2007).

Further, the potential effect of sub-vertical drains drilled from gallery (drainage gallery) on the stabilization of unstable slopes has been studied in large landslides using three-dimensional (3-D) numerical model considering hydro-geological and geo-mechanical parameter heterogeneity (Tacher et al., 2005; Matti, et. al, 2012). Although the 3-D modeling has a significant contribution on the reliability of ground movement computation, it requires a 3-D structural geological model and sophisticated transient hydro-geological model built by all available data includes boreholes, geophysics and field observations.

[revised manuscript text omitted]

(a)                         (b)

Fig. 12 (a) 14 days antecedent rainfall hyetograph prior to Amber Typhoon (1997/8/14~1997/8/28) (b) rainfall hyetograph of Amber Typhoon (1997/8/28~1997/8/29)

[Figure]

Fig. 13 Rainfall hyetograph of Toraji Typhoon (2001/7/29~2001/7/31)

**4.  Results and Discussions**

**4.1  Verification of Rainfall-Induced Seepage**

As shown in Fig. 14, the groundwater level variation of simulation without (Fig. 14(a)) and with (Fig. 14(b)) subsurface drainages remediation is tiny and agree with those of observation from *B5* monitoring station. The slight variation of groundwater level at *B5* monitoring station could be resulted from the geological feature of thin colluviums and thick fractured slate underneath in this area because this makes difficult for the soil strata to accumulate the infiltrated rainwater from long duration rainfall and to raise the groundwater level. In addition, the maximum deviation of the simulation from the observation of B5 groundwater level without and with subsurface drainage is about 0.5 m and 0.2 m respectively. These deviations are most likely caused by the simplification of 2-D numerical model which unable to capture the effect of 3-D hydrological/geological structure of soil strata.

[revised manuscript text omitted]

the calculations of *Fs* value and groundwater level are numerically dependent on the rainfall history (infiltration boundary
condition of ground surface) and the subsurface drainage (free seepage surface boundary condition of horizontal drains). In
Table 4, the increase of *Fs* values (at rainfall duration *t*=377 hr) is mainly caused by imposing a fictitious subsurface drainage
remediation on the numerical model during Amber Typhoon. As shown in Fig. 19, even though in the time duration without
precipitation (*t*=120 hr~340 hr), the fictitious drainage remains functioning and lowering the groundwater level (also increases
the *Fs* value) numerically.

Table 4 Factors of safety without and with fictitious subsurface drainages along potential sliding surfaces during Amber Typhoon in 1997

| Potential Sliding Surface (PSS) | Factor of Safety $F_S$ | | | |
| --- | --- | --- | --- | --- |
| | [1]without remediation | | [2]with fictitious remediation | |
| | Variation during rainfall | Decreasing percentage (%) | Variation during rainfall | Increasing percentage (%) |
| $1^{st}$-Potential Sliding Surface ($1^{st}$-PSS) | 1.148→1.096 | 4.53 | 1.149→1.201 | 4.53 |
| $2^{nd}$-Potential Sliding Surface (**$2^{nd}$-PSS**) | **1.317→1.263** | **4.10** | **1.351→1.403** | **3.85** |
| $3^{rd}$-Potential Sliding Surface ($3^{rd}$-PSS) | 1.250→1.210 | 3.20 | 1.304→1.409 | 8.05 |

[1]Amber Typhoon in 1997 (Fig. 12) (rainfall duration $t$=377 hr) without remediation, the subsurface drainages had not been completed yet in this duration.
[2]Amber Typhoon in 1997 (Fig. 12) (rainfall duration $t$=377 hr) with remediation, the subsurface drainages was fictitiously assigned in numerical model.
[3] $F_S$ ≥1.1 for torrential rainfall; $F_S$ ≥1.5 for ordinary time (stability criteria used in Taiwan)

Meanwhile, as shown in Fig. 20, prior to the torrential rainfall, the potential sliding surface was submerged by initial groundwater level. Subsequently at the elapsed time of typhoon rainfall, $t$=23 hr, for the occurrence of peak rainfall intensity, the groundwater level ascends for the case without drainages (Fig. 20(a)) and leads to a lower factor of safety $F_S$=1.264. On the contrary, a groundwater drawdown for the case with subsurface drainages (Fig. 20(b)) and a higher factor of safety $F_S$=1.399 can be achieved. Repeatedly, the increase of $F_S$ is due to the fact that the function of fictitious drainage imposed on the numerical model in the time duration without precipitation ($t$=0~23 hr) as shown in Fig. 19. 
[revised manuscript text omitted]
a subsurface drainage system in practice for a large scale landside.

---

## Author Comment (AC3) · 22 Apr 2016

General Comments: An interesting case study that highlights the role of drainage systems to reduce pressure heads within a landslide area is presented in the paper. The manuscript is clear and the scientific quality is generally good, although some parts lose strictness and need to be strengthened and improved. The structure of the paper can be improved, since a literature review on the specific topic (stabilization of large landslides with deep drainage interventions) lacks in the current version of the manuscript and must be added. Probably, the initial part of the manuscript (that is the general description of the landslide and the drainage works; sections 1 and 2) is too detailed and can be shortened, thus leaving space to the state-of-the-art discussion. In some parts the text is prolix, while the authors should better highlight the specific contribution of their scientific work. Response: (1) The literature review on stabilization of large landslides with deep drainage interventions has been added. Please see the introduction of the revised manuscript. (2) The initial part of the manuscript that is the general description of the landslide and the drainage works (sections 1 and 2) has been shortened and the state-of-the-art has been added and discussed. (3) The specific contribution of the scientific work has also been highlighted. Specific comments: A specific section on the subsurface hydraulics of the slope before intervention (from where the authors have started in their study) is needed, so that pre-intervention field measurements, monitoring stations, soil hydraulic parameters, along with the results of pre-intervention seepage analyses should be reported in detail. This part could be really helpful to clarify the role of drainage in the slope hydraulics. Response: The subsurface hydraulics of the slope before intervention (pre-intervention) of subsurface drainage was illustrated in the following figures and the results were also compared immediately with those of post-intervention. (1) Monitoring stations: Fig. 11 (monitoring stations B4 and B5). (2) Pre-intervention field measurements: Fig. 14 (a) B5 observation and Fig. 15 B5 monitoring station (3) Soil hydraulic parameters: Table 1 (4) Pre-intervention seepage analyses: Fig. 14 (a) B5 simulation of groundwater level variation; Fig. 17(a) pore water pressure of First potential sliding surface (1st-PPS); Fig. 20(a) groundwater levels variation; Fig. 22(a) B4 simulation of volumetric water content. The authors should better clarify the choice of assigning in their seepage analysis a "free seepage surface boundary condition" to the horizontal drainages, instead of a zero pressure head boundary condition. The current explanation of the choice is unclear. Response: (1) The horizontal drainage assigned by a "free seepage surface boundary condition" in numerical simulation is verified to be proper through a series of numerical experiments. (2) If the horizontal drainage is assigned by a "zero pressure head boundary condition with zero pressure head hp=0", some unrealistic numerical results may occur. During the seepage analysis, if a portion of horizontal drainage with zero pressure head boundary condition (pressure head hp=0) situates above the groundwater level at unsaturated zone (negative pressure head, hp<0) (or the groundwater level is lower than the horizontal drainage), eventually the horizontal drainage at unsaturated zone will numerically extract groundwater flow from saturated zone (positive pressure head, hpïĂ¿0) and this is unrealistic in engineering practice. Above explanations have been added in the relevant paragraph (Section 3.2) of the revised manuscript. How were the soil strength parameters reported in Table 2 chosen? This point requires a discussion from the authors. Response: Large quantities of field tests and laboratory tests (direct shear tests) were carried out to determine the soil strength parameters during the implementation of the subsurface drainage project. Average value of parameters was used for numerical analyses. Above discussion has been added in the relevant paragraph (Section 3.2) of the revised manuscript. Vertical scale in Figure 15 is not adequate. The authors should enlarge the scale of y-axis, so that the trend can be better appreciated. In particular, it seems that in Fig. 15a the simulation is not capable of catching the effect of cumulated rainfall, as observed in reality. Response: (1) The scale of y-axix of Figure 14 (Figure 15ïĆőFigure 14) has been enlarged. (2) As shown in Fig. 14, the maximum deviation of the simulation from the observation of B5 groundwater level without and with subsurface drainage is about 0.5 m and 0.2 m respectively. These deviations are most likely caused by the simplification of 2-D numerical model which unable to capture the effect of 3-D hydrological/geological structure of soil strata. Above comments have been given in the relevant paragraph (Section 4.1) of the revised manuscript. Table 4: it is unclear why Fs values increase during the rainfall history. This is not possible, since the effect of drainage should be a lower reduction of Fs respect to the pre-intervention situation, but not an increment. Fs could eventually increases in the long-term due to the effect of drainage, but not during a rainfall event. A comment from the authors on this point is required. Also, vertical scale in Fig. 20 should be enlarged to appreciate the trends. Response: (1) The scale of y-axix of Fig. 19 (and Fig. 18) has been enlarged (Fig. 20ïĆőFig. 19 and Fig. 19ïĆőFig. 18). (2) Numerically, the calculations of Fs value and groundwater level are largely dependent on the rainfall history (infiltration boundary condition of ground surface) and the subsurface drainage (free seepage surface boundary condition of horizontal drains). (3) In Table 4, the increase of Fs values (at rainfall duration t=377 hr) is mainly caused by imposing a fictitious subsurface drainage remediation on the numerical model during Amber Typhoon. As shown in Fig. 19, even though in the time duration without precipitation (t=120 hr 340 hr), the fictitious drainage remains functioning and lowering the groundwater level (also increases the Fs value) numerically. Related to the previous point, it is unclear why groundwater level in Fig. 21 lowers as respect to the initial groundwater level during a rainfall history. Response: Similar to the explanations in the previous point (Fig. 19), the initial groundwater level is lowered down in Fig. 20 (Fig. 21ïĆőFig. 20) due to the fact that the function of fictitious drainage imposed on the numerical model in the time duration without precipitation (t=0 23 hr) as shown in Fig. 19. Some comments and explanations have been given simultaneously for the Fs value and groundwater level in Table 4, Fig. 19 and Fig. 20. Technical comment: - section 3 is completely repeated in section 4. Please remove section 3. Response: The instruction is followed.

Please also note the supplement to this comment:
http://www.nat-hazards-earth-syst-sci-discuss.net/nhess-2015-309/nhess-2015-309-AC3-supplement.pdf
* * *
[Figure]

**Supplement:**

**Evaluating the Efficiency of Subsurface Drainages for Li-Shan Landslide in Taiwan**

Der-Guey Lin[1], Sheng-Hsiung Hung[2], Cheng-Yu Ku[3], Hsun-Chuan Chan[4]*

[1]Professor, Department of Soil and Water Conservation, National Chung-Hsing University

[2]Doctoral student, Department of Soil and Water Conservation, National Chung-Hsing University

[3]Professor, Department of Harbor and River Engineering, National Taiwan Ocean University

[4]Associate Professor, Department of Soil and Water Conservation, National Chung-Hsing University

[4]hcchan@nchu.edu.tw (*corresponding authorsd: No. 250 Kuo-Kuang Road, Taichung 402, Taiwan)

**Abstract:** This study investigates the efficiency of subsurface drainage systems includes drainage wells (vertical shaft with drainage boreholes or horizontal drains) and drainage galleries (longitudinal tunnel with sub-vertical drainage boreholes) for the slope stabilization of Li-Shan landslide in central Taiwan. The efficiency of the subsurface drainages is verified through a series of two-dimensional (2-D) rainfall induced seepage and slope stability analyses without and with subsurface drainages remediation during two typhoon events. Numerical results and monitoring data both show that the groundwater level at *B5* monitoring station with subsurface drainages remediation during Toraji Typhoon (2001) is about 40 m lower than that without remediation during Amber Typhoon (1997), and the factor of safety *Fs* of the first potential sliding surface (*1st-PSS*, the most critical potential sliding surface) is promoted simultaneously from 1.096 to 1.228 due to the function of subsurface drainage systems. In addition, the *Fs* values of the three potential sliding surfaces (*1st-PSS*, *2nd-PSS*, and *3rd-PSS*) stabilized by subsurface drainage systems are constantly maintained greater than unity ($F_S>1.0$ or $F_S \geq 1.217$) during rainfalls with return periods increases from 25 to 50 and 100 years. This demonstrates the subsurface drainage systems in Li-Shan landslide are functional and capable of accelerating the drainage of infiltration rainwater induced from high intensity and long duration rainfall and protect the slope of landslide from further deterioration. This study provides a quick computation method to evaluate the effectiveness and efficiency of a subsurface drainage system with relatively high engineering costs for a large landslide.

**Keywords:** landslides, subsurface drainage systems, drainage boreholes, drainage well, drainage gallery, potential sliding surface, factor of safety

**1. Introduction**

The Li-Shan landslide has a long history of intermittent large movements toward down slope during rainfall since 1990 and currently has been stabilized by the subsurface drainage systems implemented by Taiwan government for seven years′ duration from 1995 to 2002. The main remediation work for Li-Shan landslide was to lower the groundwater level through different subsurface drainage systems consisted of drain wells and drainage galleries. For subsurface drainage system, the effectiveness of horizontal drains on lowering the groundwater level and improving the slope stability is mainly governed by drain length, spacing, number, and installation location (Kenney et al., 1977; Prellwitz, 1978; Nonveiller, 1981; Lau and Kenney, 1984; Nakamura, 1988).

The function of horizontal drains has also been investigated by several numerical analyses. Cai et al. (1998) concluded that lengthening the horizontal drains is more effective than shortening the spacing and increasing the number of the horizontal drains in a group. Rahardjo et al. (2002, 2003, 2012) found that horizontal drains have little role in minimizing infiltration in an unsaturated residual soil slope and benefits of using horizontal drains can be obtained through the lowering of the groundwater table. In addition, the general trends of numerical results indicate the most benefit was derived from the drain located at the bottom of the slope. However, in the above analyses, the depth of horizontal drains is in comparatively shallow and unlike the deep drainage of Li-Shan landslide achieved by a series of drainage wells and drainage galleries. In addition, numerical modeling results, based on hydro-mechanically coupled 2-D distinct-element models, strongly suggest that deep drainage (drainage tunnel) was the key measure to successfully stabilize large landslides (Eberhardt et al., 2007).

Further, the potential effect of sub-vertical drains drilled from gallery (drainage gallery) on the stabilization of unstable slopes has been studied in large landslides using three-dimensional (3-D) numerical model considering hydro-geological and geo-mechanical parameter heterogeneity (Tacher et al., 2005; Matti, et. al, 2012). Although the 3-D modeling has a significant contribution on the reliability of ground movement computation, it requires a 3-D structural geological model and sophisticated transient hydro-geological model built by all available data includes boreholes, geophysics and field observations.

[revised manuscript text omitted]

(a)                         (b)

Fig. 12 (a) 14 days antecedent rainfall hyetograph prior to Amber Typhoon (1997/8/14~1997/8/28) (b) rainfall hyetograph of Amber Typhoon (1997/8/28~1997/8/29)

[Figure]

Fig. 13 Rainfall hyetograph of Toraji Typhoon (2001/7/29~2001/7/31)

**4.  Results and Discussions**

**4.1  Verification of Rainfall-Induced Seepage**

As shown in Fig. 14, the groundwater level variation of simulation without (Fig. 14(a)) and with (Fig. 14(b)) subsurface drainages remediation is tiny and agree with those of observation from *B5* monitoring station. The slight variation of groundwater level at *B5* monitoring station could be resulted from the geological feature of thin colluviums and thick fractured slate underneath in this area because this makes difficult for the soil strata to accumulate the infiltrated rainwater from long duration rainfall and to raise the groundwater level. In addition, the maximum deviation of the simulation from the observation of B5 groundwater level without and with subsurface drainage is about 0.5 m and 0.2 m respectively. These deviations are most likely caused by the simplification of 2-D numerical model which unable to capture the effect of 3-D hydrological/geological structure of soil strata.

[revised manuscript text omitted]

the calculations of *Fs* value and groundwater level are numerically dependent on the rainfall history (infiltration boundary
condition of ground surface) and the subsurface drainage (free seepage surface boundary condition of horizontal drains). In
Table 4, the increase of *Fs* values (at rainfall duration *t*=377 hr) is mainly caused by imposing a fictitious subsurface drainage
remediation on the numerical model during Amber Typhoon. As shown in Fig. 19, even though in the time duration without
precipitation (*t*=120 hr~340 hr), the fictitious drainage remains functioning and lowering the groundwater level (also increases
the *Fs* value) numerically.

Table 4 Factors of safety without and with fictitious subsurface drainages along potential sliding surfaces during Amber Typhoon in 1997

| Potential Sliding Surface (PSS) | Factor of Safety $F_S$ | | | |
| --- | --- | --- | --- | --- |
| | [1]without remediation | | [2]with fictitious remediation | |
| | Variation during rainfall | Decreasing percentage (%) | Variation during rainfall | Increasing percentage (%) |
| $1^{st}$-Potential Sliding Surface ($1^{st}$-PSS) | 1.148→1.096 | 4.53 | 1.149→1.201 | 4.53 |
| $2^{nd}$-Potential Sliding Surface (**$2^{nd}$-PSS**) | **1.317→1.263** | **4.10** | **1.351→1.403** | **3.85** |
| $3^{rd}$-Potential Sliding Surface ($3^{rd}$-PSS) | 1.250→1.210 | 3.20 | 1.304→1.409 | 8.05 |

[1]Amber Typhoon in 1997 (Fig. 12) (rainfall duration $t$=377 hr) without remediation, the subsurface drainages had not been completed yet in this duration.
[2]Amber Typhoon in 1997 (Fig. 12) (rainfall duration $t$=377 hr) with remediation, the subsurface drainages was fictitiously assigned in numerical model.
[3] $F_S$ ≥1.1 for torrential rainfall; $F_S$ ≥1.5 for ordinary time (stability criteria used in Taiwan)

Meanwhile, as shown in Fig. 20, prior to the torrential rainfall, the potential sliding surface was submerged by initial groundwater level. Subsequently at the elapsed time of typhoon rainfall, $t$=23 hr, for the occurrence of peak rainfall intensity, the groundwater level ascends for the case without drainages (Fig. 20(a)) and leads to a lower factor of safety $F_S$=1.264. On the contrary, a groundwater drawdown for the case with subsurface drainages (Fig. 20(b)) and a higher factor of safety $F_S$=1.399 can be achieved. Repeatedly, the increase of $F_S$ is due to the fact that the function of fictitious drainage imposed on the numerical model in the time duration without precipitation ($t$=0~23 hr) as shown in Fig. 19. 
[revised manuscript text omitted]
a subsurface drainage system in practice for a large scale landside.